# LQ-LoRA: Low-rank Plus Quantized Matrix Decomposition for Efficient Language Model Finetuning

**Han Guo**[†⋆]  **Philip Greengard**[‡]  **Eric P. Xing**[†◇]  **Yoon Kim**[⋆]

[†]Carnegie Mellon University, [‡]Columbia University
[◇]Mohamed bin Zayed University of Artificial Intelligence, Petuum Inc.
[⋆]Massachusetts Institute of Technology
hanguo@cs.cmu.edu, pg2118@columbia.edu, epxing@cs.cmu.edu, yoonkim@mit.edu

## ABSTRACT

We propose a simple approach for memory-efficient adaptation of pretrained language models. Our approach uses an iterative algorithm to decompose each pretrained matrix into a high-precision low-rank component and a memory-efficient quantized component. During finetuning, the quantized component remains fixed and only the low-rank component is updated. We present an integer linear programming formulation of the quantization component which enables dynamic configuration of quantization parameters (e.g., bit-width, block size) for each matrix given an overall target memory budget. We further explore a data-aware version of the algorithm which uses an approximation of the Fisher information matrix to weight the reconstruction objective during matrix decomposition. Experiments on finetuning RoBERTa and LLaMA-2 (7B and 70B) demonstrate that our low-rank plus quantized matrix decomposition approach (LQ-LoRA) outperforms strong QLoRA and GPTQ-LoRA baselines and enables aggressive quantization to sub-3 bits with only minor performance degradations. When finetuned on a language modeling calibration dataset, LQ-LoRA can also be used for model compression; in this setting our 2.75-bit LLaMA-2-70B model (which has 2.85 bits on average when including the low-rank components and requires 27GB of GPU memory) performs respectably compared to the 16-bit baseline.[1]

## 1 INTRODUCTION

Despite the increased availability of large language models (LLMs) and their pretrained parameters (Zhang et al., 2022; Scao et al., 2022; Touvron et al., 2023a;b), their sheer size makes them expensive to adapt to new datasets via full finetuning. This is particularly unideal since a small amount of supervised finetuning on instruction following data has been shown to be an effecive approach for learning interactive agents that can follow general instructions (Wang et al., 2023; Taori et al., 2023; Team, 2023; Zhou et al., 2023), and moreover, LLMs finetuned via reinforcement learning with human feedback (Ouyang et al., 2022) represent some of the most capable AI systems that exist today (OpenAI, 2023; Bubeck et al., 2023). Improving the memory-efficiency of LLM finetuning thus remains a key step in widening the scope of problems to which LLMs can be practically applied.

One promising framework for memory-efficient LLM adaptation is through parameter-efficient finetuning methods, which typically learn a smaller finetunable *extension* to the base pretrained model (see Ding et al. (2023) for a survey). These methods can reduce the amount of memory required for finetuning as the pretrained parameters remain fixed—thus reducing the need to allocate memory for storing gradients and optimizer states for these parameters—while the number of new parameters to be optimized is a fraction of the fixed parameters. Of the many existing parameter-efficient finetuning methods, low-rank adaptation (LoRA; Hu et al., 2022) has emerged as a popular technique for efficient LLM adaptation. In LoRA, the pretrained model's weight matrix $\mathbf{W}$ is reparameterized as $\mathbf{W} + \mathbf{L}_1\mathbf{L}_2$, and only $\mathbf{L}_1$ and $\mathbf{L}_2$ are finetuned. Recent works have improved the memory-efficiency of LoRA further by applying it to a quantized pretrained model, i.e., using the reparameterization $q(\mathbf{W}) + \mathbf{L}_1\mathbf{L}_2$ where $q(\cdot)$ is some quantization function (Dettmers et al., 2023a; Chai et al., 2023).

---

[1]Our code and models are available at https://github.com/HanGuo97/lq-lora. This work was completed while Han Guo was a visiting student at MIT.

In LoRA, $\mathbf{L}_2$ is initialized to $\mathbf{0}$ to ensure that the model output is the same as the pretrained model at the beginning of finetuning (i.e., $\mathbf{X}(\mathbf{W} + \mathbf{L}_1\mathbf{L}_2) = \mathbf{X}\mathbf{W}$). However, if the pretrained matrices are quantized to the extent where there is substantial quantization error (which has been empirically found to occur at sub-4-bit regimes), zero initialization may not be optimal since $q(\mathbf{W}) + \mathbf{L}_1\mathbf{L}_2 \neq \mathbf{W}$. In this paper, we exploit the fact that LoRA only performs low-rank updates to the quantized model to derive an initialization scheme that takes the quantization error into account. We use an iterative algorithm similar to those used in the robust PCA literature (Wright et al., 2009; Candès et al., 2011; Zhou & Tao, 2011) to decompose $\mathbf{W}$ such that $\mathbf{W} \approx \mathbf{Q} + \mathbf{L}_1\mathbf{L}_2$. Here $\mathbf{Q}$ is the quantized component which remains fixed and $\mathbf{L}_1\mathbf{L}_2$ is the low-rank component. During adaptation only $\mathbf{L}_1$ and $\mathbf{L}_2$ (which captures the high-variance subspaces of $\mathbf{W}$) are finetuned. Instead of applying the same quantization configuration to all layers, we use integer linear programming to find a mixed quantization strategy that allows for the assignment of different configurations (bits, block size, etc.) to each matrix given an overall target bit rate. Finally, we explore a data-aware version of the algorithm which modifies the decomposition objective with an approximation of the Fisher information matrix obtained from calibration samples.

We apply LQ-LoRA to adapt RoBERTa (Liu et al., 2019) and LLaMA-2 (Touvron et al., 2023b) models and find that it can meaningfully improve upon strong QLoRA (Dettmers et al., 2023a) and GPTQ-LoRA (Frantar et al., 2022; Chai et al., 2023) baselines while enabling users to flexibly set a target memory budget. LQ-LoRA can also be applied on standard language modeling datasets to serve as a weight-only post-training quantization (PTQ) method. In this setting we find that we are able to compress LLaMA-2-70B to 2.85 bits with only a small perplexity degradation.

## 2 BACKGROUND

### 2.1 LOW-RANK ADAPTATION OF LARGE LANGUAGE MODELS

Low-rank adaptation of large language models (LoRA; Hu et al., 2022) has emerged as a simple but effective approach for reducing the memory footprint during LLM finetuning. Given a matrix $\mathbf{W} \in \mathbb{R}^{d \times k}$ of a pretrained linear layer, LoRA initializes two matrices $\mathbf{L}_1 \in \mathbb{R}^{d \times r}, \mathbf{L}_2 \in \mathbb{R}^{r \times k}$ with $r < \min(d, k)$, where $\mathbf{L}_1$ is initialized to Gaussian noise and $\mathbf{L}_2$ is initialized to $\mathbf{0}$ (in order to ensure that $\mathbf{L}_1\mathbf{L}_2 = \mathbf{0}$ at the start of training). LoRA then reparameterizes the linear layer as $\mathbf{X}(\mathbf{W} + \mathbf{L}_1\mathbf{L}_2)$ (here $\mathbf{X}$ is the previous layer's activation), and only finetunes $\mathbf{L}_1$ and $\mathbf{L}_2$ during language model adaptation. (The bias vector is omitted for brevity.) LoRA is more memory-efficient than full finetuning as there is no need to allocate GPU memory for the gradients and the associated optimizer states (e.g., the momentum and variance statistics in Adam (Kingma & Ba, 2015)) for $\mathbf{W}$.

### 2.2 WEIGHT QUANTIZATION OF LARGE LANGUAGE MODELS

Standard round-to-nearest (RTN) quantization, which quantizes/dequantizes a block of weights as $\mathbf{u} \approx s \times \text{clamp}\left(\left\lfloor \frac{1}{s}\mathbf{u} \right\rceil; -2^{b-1}, 2^{b-1} - 1\right)$ with scaling factor $s = \frac{\max(|\mathbf{u}|)}{2^{b-1}-1}$ and bit size $b$, has been shown to be effective for quantizing a pretrained LLM's weights to 8-bits (Yao et al., 2022). However, (sub) 4-bit quantization has been empirically found to be difficult with RTN, and recent methods generally employ a data-aware strategy which uses calibration samples to obtain better weight quantization (Frantar et al., 2022; Dettmers et al., 2022; Xiao et al., 2022; Kim et al., 2023b; Lin et al., 2023; Dettmers et al., 2023b; Shao et al., 2023, *inter alia*).

Our approach relies on the recently proposed NormalFloat (NF) quantization scheme (Dettmers et al., 2023a), which exploits the fact that the distribution of the weights of a trained model is approximately Gaussian. Following the presentation from Yoshida (2023), NF quantization calculates $2^{b-1}$ evenly-spaced values from $[\delta, \frac{1}{2}]$, and $2^{b-1} + 1$ evenly-spaced values from $[\frac{1}{2}, 1 - \delta]$, where $\delta = \frac{1}{2}(\frac{1}{30} + \frac{1}{32})$. This results in $2^b$ probability values $[p_1, \ldots, p_{2^b}]$ where $p_1 = \delta, p_{2^{b-1}} = \frac{1}{2}$, and $p_{2^b} = 1 - \delta$. These probabilities are converted into quantiles $[q_1, \ldots, q_{2^b}]$ where $q_i = \Phi^{-1}(p_i)$ is the Gaussian quantile for $p_i$, and these quantiles are normalized to $[-1, 1]$ by $\tilde{q}_i = \frac{q_i}{q_{2^b}}$. Then, given a block of weights $\mathbf{u} = [u_1, \ldots, u_B]$ and the absmax value $s = \max(|\mathbf{u}|)$ for that block, the weights $u_j$ in this block are quantized to the nearest quantile $c_j$, i.e., $c_j = \arg\min_{i \in \{1, \ldots, 2^b\}} \left| \tilde{q}_i - \frac{u_j}{s} \right|$.

For a $d \times k$ matrix there are $\frac{dk}{B}$ blocks, and hence storing the absmax values $s$ for each block could become substantial with small block sizes. Dettmers et al. (2023a) thus employ a double quantization strategy where the set of absmax values $[s_1, \ldots, s_{\frac{dk}{B}}]$ for a given matrix are quantized again via RTN. Based on this quantization scheme, Dettmers et al. (2023a) propose QLoRA, which

performs NF quantization to 4 bits on the pretrained LLM, and learns low-rank updates. QLoRA has been found to be competitive with full finetuning across a number of benchmarks, and thus serves as the main baseline of the present work.

## 3 METHOD: LQ-LoRA

Our approach relies on a simple factorization scheme which decomposes each pretrained matrix into a low-rank matrix plus a quantized matrix (§3.1), where only the low-rank component is adapted during finetuning. In §3.2 we explore a mixed quantization strategy via integer linear programming to allow for dynamic quantization across layers given a target average bit rate. We further consider a data-aware version of LQ-LoRA by using the empirical Fisher information matrix to weight the reconstruction objective during matrix factorization (§3.3).

### 3.1 LOW-RANK PLUS QUANTIZED MATRIX DECOMPOSITION

As noted in §2.1, LoRA reparameterizes a pretrained matrix as $\mathbf{W}$ as $\mathbf{W} + \mathbf{L}_1\mathbf{L}_2$ and initializes $\mathbf{L}_1$ from a Gaussian and $\mathbf{L}_2$ to $\mathbf{0}$ before finetuning. While this ensures that the model output is exactly the same as before reparameterization at the start of finetuning, it may present an issue when working with a quantized version of $\mathbf{W}$ since we could have $\|\mathbf{W} - \mathrm{Quantize}(\mathbf{W})\|_F \gg 0$ when quantizing to low bits. This initialization moreover does not take into account $\mathbf{W}$'s structure when deciding on which subspaces to adapt. We approach this problem from the perspective of matrix factorization where we are interested factorizing the original matrix into an easily quantizable component and a low-rank component that captures high-variance directions,

$$\underset{\mathbf{Q},\mathbf{L}_1,\mathbf{L}_2}{\arg\min} \|\mathbf{W} - (\mathbf{Q} + \mathbf{L}_1\mathbf{L}_2)\|_F, \quad \text{where } \mathbf{Q} \in \mathbb{Q}_b^{d \times k}, \mathbf{L}_1 \in \mathbb{R}^{d \times r}, \mathbf{L}_2 \in \mathbb{R}^{r \times k}. \tag{1}$$

Here $\mathbb{Q}_b^{d \times k} \subset \mathbb{R}^{d \times k}$ is the set of matrices that are losslessly NF-quantizable to $b$-bits. This optimization problem is similar to the one faced in robust principal components analysis (RPCA; Wright et al., 2009; Candès et al., 2011), which aims to decompose a matrix $\mathbf{W}$ into $\mathbf{L} + \mathbf{S}$ where $\mathbf{L}$ is low-rank and $\mathbf{S}$ is *sparse*. Following iterative algorithms which have been shown to be effective for RCPA (Lin et al., 2010; Zhou & Tao, 2011), we approximately solve Eq. 1 via alternating between optimizing $\mathbf{L}_1\mathbf{L}_2$, and $\mathbf{Q}$:[2]

$$
\begin{aligned}
\mathbf{L}_1^{(t)}, \mathbf{L}_2^{(t)} &\leftarrow \mathrm{SVD}(\mathbf{W} - \mathbf{Q}^{(t-1)}, r), && = \underset{\mathrm{rank}(\mathbf{L}) \leq r}{\arg\min} \|\mathbf{W} - (\mathbf{Q}^{(t-1)} + \mathbf{L})\|_F, \\
\mathbf{Q}^{(t)} &\leftarrow \mathrm{Quantize}(\mathbf{W} - \mathbf{L}_1^{(t)}\mathbf{L}_2^{(t)}), && \approx \underset{\mathbf{Q} \in \mathbb{Q}_b^{d \times k}}{\arg\min} \|\mathbf{W} - (\mathbf{Q} + \mathbf{L}_1^{(t)}\mathbf{L}_2^{(t)})\|_F,
\end{aligned}
\tag{2}
$$

where $\mathbf{Q}^{(0)}$ is initialized to $\mathbf{0}$. Unlike (some) RPCA algorithms for which theoretical convergence guarantees can be obtained (Ma & Aybat, 2018), the above algorithm is heuristic. We thus employ a simple stopping criterion where we keep track of the error $\|\mathbf{W} - (\mathbf{Q}^{(t)} + \mathbf{L}_1^{(t)}\mathbf{L}_2^{(t)})\|_F$ and terminate the algorithm if the error increases. The iterative decomposition algorithm is shown in Algorithm 2.[3] Each step of the algorithm (i.e., randomized SVD followed by quantization) takes a few seconds on a modern GPU for a $4096 \times 4096$ matrix.

**Preliminary experiments.** In Figure 1 (left) we show the decomposition error $\|\mathbf{W} - (\mathbf{Q} + \mathbf{L}_1\mathbf{L}_2)\|_F$ for a few layers of LLaMA-2-7B as a function of the number of steps. We find that our algorithm, while heuristic, is empirically effective. In Figure 1 (center) we show the quantization error for 3-bit NF quantization for all matrices, while in Figure 1 (right) we show the corresponding error for LQ decomposition. For both approaches we find that the value and output projection matrices become harder to quantize at deeper layers, while the key and query matrices become easier; however, our LQ decomposition is able to improve upon vanilla quantization for all layers.

---

[2] In practice we use randomized SVD instead of full SVD, which significantly reduced runtime for the SVD portion of the algorithm without much deterioration in performance.

[3] Despite the simplicity of our approach, we are not aware of prior work on low-rank plus quantized matrix decomposition, except for a recent preprint which proposes to perform SVD on the residuals $\mathbf{E} = \mathbf{W} - \mathrm{Quantize}(\mathbf{W})$ to correct for errors after quantization (Yao et al., 2023). This approach can be seen as performing a single step of the iterative algorithm with the initialization $\mathbf{Q}^{(0)} = \mathrm{Quantize}(\mathbf{W})$. In our experiments we did not observe significant differences in performance when we initialized $\mathbf{Q}^{(0)}$ to $\mathrm{Quantize}(\mathbf{W})$.

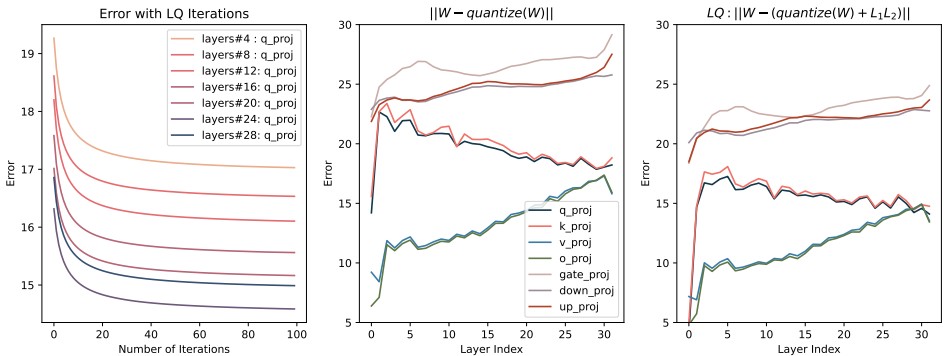

**Figure 1:** (Left) The decomposition error $\|\mathbf{W} - (\mathbf{Q} + \mathbf{L}_1\mathbf{L}_2)\|_F$ for the query projection matrices for different layers of LLaMA-2-7B as a function of the number of LQ steps. (Center) Quantization error for NF-3 quantization for all layers. (Right) LQ decomposition error for 3-bit quantization with rank = 64. LQ decomposition results in less quantization error.

## 3.2 MIXED-CONFIGURATION QUANTIZATION VIA AN INTEGER LINEAR PROGRAM

LQ-LoRA uses the NormalFloat (NF) quantization scheme from Dettmers et al. (2023a) to quantize the residual $\mathbf{Q}$ at each time step. NF-quantization has several parameters that affect the overall compression rate such as the number of quantile bins, number of blocks, and bits for double quantization. In this paper we work with slightly different variant which quantizes a matrix $\mathbf{A}$ via the following:

$$\widehat{\mathbf{A}}, \mathbf{s} = \text{Quantize-NF}\left(\mathbf{A}, b_0, B_0\right), \quad \widehat{\mathbf{s}}, \mathbf{v} = \text{Quantize-INT}\left(\mathbf{s}, b_1, B_1\right), \quad \widehat{\mathbf{v}} = \text{cast}\left(\mathbf{v}, b_2\right).$$

Concretely, we first apply NF-quantization with bit size $b_0$ and bucket size $B_0$ to obtain the quantized matrix $\widehat{\mathbf{A}}$ and the absmax values for each block $\mathbf{s} = [s_1, \ldots, s_{\frac{dk}{B_0}}]$ (see §2.2). These absmax values are further quantized to $b_1$ bits via uniform integer quantization with bucket size $B_1$ to obtain the quantized vector $\widehat{\mathbf{s}}$, along with the absmax values for $\mathbf{s}$, i.e., $\mathbf{v} = [v_1, \ldots v_{\frac{dk}{B_0 B_1}}]$.[4] Finally, we cast $\mathbf{v}$ to $b_2$ bits to obtain $\widehat{\mathbf{s}_1}$.[5] Dequantization, which is needed on the fly for finetuning and inference, simply reverses this process.

This quantization scheme requires storing $\widehat{\mathbf{A}}, \widehat{\mathbf{s}}, \widehat{\mathbf{v}}$ to represent $\mathbf{A}$. We can thus quantify the storage cost (number of bits) for storing $\mathbf{A}$ given a configuration $c = (b_0, b_1, b_2, B_0, B_1)$ as

$$\text{storage}(\mathbf{A}, c) = \text{sizeof}(\mathbf{A}) \cdot \left(b_0 + \frac{b_1}{B_0} + \frac{b_2}{B_0 \cdot B_1}\right). \tag{3}$$

The original NF-4 double quantization is a special case with $c_{\text{NF4}} = (4, 8, \text{fp32}, 64, 256)$ and $\text{storage}(\mathbf{A}, c_{\text{NF4}}) = 4.127 \cdot \text{sizeof}(\mathbf{A})$, i.e., NF-4 requires on average 4.127 bits per parameter.

**Dynamic quantization configurations.** Prior works on quantizing LLMs have generally focused on applying the same quantization strategy to each matrix, which cannot adapt to users' varying resource constraints and moreover may be suboptimal given that some matrices may be harder to quantize than others. We explore a mixed-precision quantization strategy based on integer linear programming (Yao et al., 2021; Tang et al., 2022; Kundu et al., 2022), which allows for the allocation of different configurations to each matrix given a user-defined target target bit rate.

Let $c = (b_0, b_1, b_2, B_0, B_1)$ be the configuration parameters and further let $\mathcal{C}$ be the set of possible configurations which is specified by the user (see Table 5 for the settings we consider in this work).

Letting $\{\mathbf{W}^{(i)}\}_{i \in [N]}$ be the set of $N$ matrices in an LM, our goal is to find an assignment matrix $\mathbf{X} \in \{0, 1\}^{N \times |\mathcal{C}|}$ that minimizes the Frobenius norm between the matrices before and after low-rank plus quantized decomposition, while respecting a target memory budget. One way to approach this

---

[4]I.e., given $v_1 = \text{absmax}([s_1, \ldots, s_{B_1}])$ for a group of size $B_1$ we have $\hat{s}_i = \text{clamp}\left(\lfloor \frac{s_i}{v_1} \rceil; 0, 2^{b_1-1}\right)$.

[5]This approach deviates from the original approach in that we use integer quantization on $\mathbf{s}$ as opposed to FP8 (which did not affect results), and we cast $\mathbf{v}$ to lower precision (which led to negligible increase in error).

**Algorithm 1** LQ-LoRA (Section 3)

**Input:** $\{\mathbf{W}^{(i)}\}_{i\in[N]}$: Parameters
$\{\mathbf{F}^{(i)}\}_{i\in[N]}$: Fisher information (optional)
$\mathcal{C}$: List of quantization configurations
$r$: LoRA rank
$B_Q$: Quantization budget
# get quantization configurations (Section 3.2).
$\{c^{(i)}\} \leftarrow \text{GetConfig}(\{\mathbf{W}^{(i)}, \mathbf{F}^{(i)}\}, \mathcal{C}, r, B_Q)$
**for** $i \leftarrow 1$ to $N$ **do**
    # matrix decomposition (Section 3.1).
    $\mathbf{Q}^{(i)}, \mathbf{L}_1^{(i)}, \mathbf{L}_2^{(i)}, \epsilon \leftarrow \text{LQ}(\mathbf{W}^{(i)}, \mathbf{F}^{(i)}, c^{(i)}, r)$
**Output:** $\{\mathbf{Q}^{(i)}, \mathbf{L}_1^{(i)}, \mathbf{L}_2^{(i)}\}_{i\in[N]}$

---

**Algorithm 2** LQ (Section 3.1)

**Input:** $\mathbf{W}$: Input weight matrix
$\mathbf{F}$: Fisher information (optional)
$c$: Quantization configuration
$r$: Target rank
Initialize $\mathbf{Q} \leftarrow \mathbf{0}$ and $\epsilon_0 \leftarrow \infty$
**for** $t \leftarrow 1$ to $T$ **do**
    $\mathbf{L}_1, \mathbf{L}_2 \leftarrow \text{Factorize}(\mathbf{W} - \mathbf{Q}, \mathbf{F}, r)$
    $\mathbf{Q} \leftarrow \text{Quantize}\,(\mathbf{W} - \mathbf{L}_1\mathbf{L}_2, c)$
    **if** $\mathbf{F}$ is None **then**
        $\epsilon_t \leftarrow \|\mathbf{W} - (\mathbf{Q} + \mathbf{L}_1\mathbf{L}_2)\|_F$
    **else**
        # weighted error (Section 3.3).
        $\epsilon_t \leftarrow \left\| \sqrt{\mathbf{F}} \odot (\mathbf{W} - (\mathbf{Q} + \mathbf{L}_1\mathbf{L}_2)) \right\|_F$
    **if** $\epsilon_t > \epsilon_{t-1}$ **then** break
**Output:** $\mathbf{Q}, \mathbf{L}_1, \mathbf{L}_2, \epsilon_t$

---

**Algorithm 3** GetConfig (Section 3.2)

**Input:** $\{\mathbf{W}^{(i)}\}_{i\in[N]}$: Parameters
$\{\mathbf{F}^{(i)}\}_{i\in[N]}$: Fisher information (optional)
$\mathcal{C}$: List of quantization configurations
$r$: Target rank
$B_Q$: Quantization budget
$\mathbf{E}, \mathbf{S} \leftarrow \text{zeros}(N, |\mathcal{C}|)$ # initialize error and storage
**for** $i \leftarrow 1$ to $N$ **do**
    **for** $c \in \mathcal{C}$ **do**
        $\mathbf{Q}^{(i)}, \mathbf{L}_1^{(i)}, \mathbf{L}_2^{(i)}, \epsilon \leftarrow \text{LQ}(\mathbf{W}^{(i)}, \mathbf{F}^{(i)}, c, r)$
        $\mathbf{E}[i, c] \leftarrow \epsilon^2$
        $\mathbf{S}[i, c] \leftarrow \text{storage}(\mathbf{W}^{(i)}, c)$
# get optimal configuration given budget with ILP.
$\{c^{(i)}\}_{i\in[N]} \leftarrow \text{ILPSolve}(\mathbf{S}, \mathbf{E}, B_Q)$
**Output:** $\{c^{(i)}\}_{i\in[N]}$

---

**Algorithm 4** Factorize (Section 3.3)

**Input:** $\mathbf{A}$: Input matrix
$\mathbf{F}$: SVD weighting matrix (optional)
$r$: Target rank
**if** $\mathbf{F}$ is None **then**
    # (randomized) SVD with target rank $r$
    $[\mathbf{U}, \boldsymbol{\Sigma}, \mathbf{V}^\top] \leftarrow \text{SVD}(\mathbf{A}, r)$
    $\mathbf{L}_1 \leftarrow \mathbf{U}\sqrt{\boldsymbol{\Sigma}}, \; \mathbf{L}_2 \leftarrow \sqrt{\boldsymbol{\Sigma}}\mathbf{V}^\top$
**else**   # weighted SVD (Section 3.3).
    $\mathbf{D}_{\text{row}} \leftarrow \text{RowAverage}(\mathbf{F})$
    $\mathbf{D}_{\text{col}} \leftarrow \text{ColAverage}(\mathbf{F})$
    $[\mathbf{U}, \boldsymbol{\Sigma}, \mathbf{V}^\top] \leftarrow \text{SVD}(\mathbf{D}_{\text{row}}\mathbf{A}\mathbf{D}_{\text{col}}, r)$
    $\mathbf{L}_1 \leftarrow \mathbf{D}_{\text{row}}^{-1}\mathbf{U}\sqrt{\boldsymbol{\Sigma}}, \; \mathbf{L}_2 \leftarrow \sqrt{\boldsymbol{\Sigma}}\mathbf{V}^\top\mathbf{D}_{\text{col}}^{-1}$
**Output:** $\mathbf{L}_1, \mathbf{L}_2$

---

optimization problem is through the following integer linear program,[6]

$$\min_{\mathbf{X}\in\{0,1\}^{N\times|\mathcal{C}|}} \quad \sum_{i\in[N]}\sum_{c\in\mathcal{C}} \text{error}(\mathbf{A}^{(i)}, c) \cdot \mathbf{X}[i, c],$$

$$\text{subject to} \quad \sum_{i\in[N]}\sum_{c\in\mathcal{C}} \text{storage}(\mathbf{A}^{(i)}, c) \cdot \mathbf{X}[i, c] \leq \text{budget},$$

$$\sum_{c\in\mathcal{C}} \mathbf{X}[i, c] = 1, \quad \forall i \in [N].$$

Here $\text{error}(\mathbf{W}^{(i)}, c) = \|\mathbf{W}^{(i)} - (\mathbf{Q} + \mathbf{L}_1\mathbf{L}_2)\|_F^2$ is the reconstruction error after running the iterative algorithm from Sec. 3.1 where the Quantize function uses configuration $c$. To approximately solve this ILP we pre-compute the errors for all matrices and quantization configurations ($|\mathcal{C}| = 3^5$) and use an off-the-shelf solver,[7] as shown in Algorithm 3. The pre-computation is a one-time process and takes a few hours when parallelized across four A100 GPUs for LLaMA-2-7B. Once the (approximately) optimal configuration $c^{(i)}$ is found, we apply the decomposition on $\mathbf{W}^{(i)}$ one more time with $c^{(i)}$ to obtain the final matrices $\mathbf{Q}^{(i)}, \mathbf{L}_1^{(i)}, \mathbf{L}_2^{(i)}$ for $i \in [N]$ (see Algorithm 1).

**Implementation.** We use `__torch_dispatch__` to duck-type `torch.Tensor` and overload PyTorch operations to perform just-in-time dequantization. We then use PyTorch's compiler to compile the bits-unpacking, dequantization, other linear algebra operations. For batch size $> 1$, this PyTorch-based implementation (followed by compilation) was as fast as some custom CUDA implementations such as `bitsandbytes`.[8] Further details and speed comparisons are given in Appendix B.

### 3.3 DATA-AWARE MATRIX DECOMPOSITION VIA FISHER-WEIGHTED SVD

The decomposition objective considered in §3.1 is data-agnostic insofar as it treats each entry of $\mathbf{W}$ as equally important for reconstruction during factorization. Following recent works which demon-

---

[6]Here we overload $c$ to refer to both its tuple representation $c \in \mathcal{C}$ and its index representation $c \in [|\mathcal{C}|]$.
[7]https://www.gurobi.com/
[8]https://github.com/TimDettmers/bitsandbytes

strate the importance of using calibration data for quantizating LLMs (Frantar et al., 2022; Lin et al., 2023; Kim et al., 2023b), we next consider a data-aware version of the approach by using a diagonal approximation of the Fisher information matrix to weight the reconstruction objective. The (diagonal of the) empirical Fisher information matrix measures how sensitive the model's output is to a perturbation of each parameter, and has previously been exploited to improve low-rank compression (Hsu et al., 2022) and quantization (Kim et al., 2023b) of pretrained language models. When applied to the LQ decomposition algorithm from §3.1, this results in the following weighted SVD problem: given $\mathbf{E} := \mathbf{W} - \mathbf{Q}$ and weighting matrix $\mathbf{F}$, we must find matrices $\mathbf{L}_1 \in \mathbb{R}^{d \times r}, \mathbf{L}_2 \in \mathbb{R}^{r \times k}$ that form the best rank-$r$ approximation, $\mathbf{L}_1, \mathbf{L}_2 = \arg\min_{\mathbf{L}_1, \mathbf{L}_2} \left\| \sqrt{\mathbf{F}} \odot (\mathbf{E} - \mathbf{L}_1 \mathbf{L}_2) \right\|_F^2$, where $\odot$ is the Hadamard product. Unliked its unweighted counterpart, this problem is in general intractable (and in fact NP-hard; Razenshteyn et al., 2016) and is typically addressed through approximate methods (Srebro & Jaakkola, 2003; Li et al., 2016; Tuzhilina & Hastie, 2021). However, if we assume that either rows or columns of the weight matrix $\mathbf{F}$ have identical values, the above problem can be solved exactly by standard SVD,

$$\mathbf{U}, \mathbf{\Sigma}, \mathbf{V}^\top \leftarrow \text{SVD}(\mathbf{D}_{\text{row}} \mathbf{A} \mathbf{D}_{\text{col}}), \quad \mathbf{L}_1 \leftarrow \mathbf{D}_{\text{row}}^{-1} \mathbf{U} \sqrt{\mathbf{\Sigma}}, \quad \mathbf{L}_2 \leftarrow \sqrt{\mathbf{\Sigma}} \mathbf{V}^\top \mathbf{D}_{\text{col}}^{-1}. \quad (4)$$

where $\mathbf{D}_{\text{row}}$ and $\mathbf{D}_{\text{col}}$ are diagonal matrices that consist of row- and column-means of $\sqrt{\mathbf{F}}$. Please see Algorithm 4 and §A for details. While the homogenous row/column assumption clearly does not hold for $\mathbf{F}$, we found this approach to work well in practice.[9] We note that this approximation is a simple extension of Hsu et al. (2022) who use $\mathbf{D}_{\text{row}}$ but not $\mathbf{D}_{\text{col}}$ in their weighted SVD (we found that using both the row- and column-averages performed slightly better).

**Discussion.** This data-aware version of LQ-LoRA requires backpropagating through the pretrained LM to obtain the Fisher matrices. We note however, that we compute $\{\mathbf{F}^{(i)}\}_{i \in [N]}$ based on some generic text data to obtain the LQ-LoRA initializations, and use the *same* initialization for different downstream tasks. This makes the data-aware approach practical, as the Fisher computaton and the matrix decomposition needs to performed only once (as in the non-data-aware version).

## 4 EMPIRICAL STUDY

We conduct experiments with LQ-LoRA across three settings: (1) continual language modeling on C4 training data, (2) instruction tuning on the OpenAssistant dataset (Köpf et al., 2023), (3) and finetuning on GLUE (Wang et al., 2018). For (1) and (2) we work with LLaMA-2 models (Touvron et al., 2023b), while for (3) we use RoBERTa-Large (Liu et al., 2019). Our setup closely follows the setup from Dettmers et al. (2023a). The Fisher-weighted version of LQ-LoRA uses randomly sampled sequences from the C4 training set, where for RoBERTa-Large we employ the masked language modeling objective (also on C4) to obtain the Fisher matrix.

**Baselines.** Our main baselines include QLoRA (Dettmers et al., 2023a) and GPTQ-LoRA. Both approaches perform PTQ on the pretrained model before learning low-rank updates to the quantized model for adaptation; QLoRA uses NF-quantization, while GPTQ-LoRA uses approximate second-order information to solve for $\arg\min_{\hat{\mathbf{W}} \in \mathbb{Q}_b^{d \times k}} \|\mathbf{X}\mathbf{W} - \mathbf{X}\,\hat{\mathbf{W}}\|_F$ (Frantar & Alistarh, 2022; Frantar et al., 2022). As the original papers were applied on top of LLaMA-1 (Touvron et al., 2023a), for fair comparison we reimplement these baselines on top of LLaMA-2. We follow Dettmers et al. (2023a) use rank = 64 for our main experiments, and ablate on the rank in our analysis section.

**Evaluation.** To evaluate models trained on C4, we use three metrics: perplexity on C4 validation, perplexity on WikiText-2 (Merity et al., 2016), and 5-shot MMLU accuracy (Hendrycks et al., 2021). For instruction-tuned models,[10] we use a Vicuna-style automatic evaluation (Team, 2023). This involves asking GPT-4 to make pairwise comparisons between its outputs and those of GPT-3.5 (with the possibility of a tie) over 80 curated questions. We chose this evaluation scheme over the 10-point rating system, following the recommended setup from Dettmers et al. (2023a).[11] For the GLUE benchmark, we show the average metrics across all tasks.

---

[9]In preliminary experiments we also explored a version of data-aware LQ-LoRA where we approximately minimized $\|\mathbf{X}(\mathbf{W} - (\mathbf{Q} + \mathbf{L}_1 \mathbf{L}_2))\|_F$ using activations $\mathbf{X}$ from calibration data, instead of the Fisher information matrix. However we found this to underperform the Fisher approach.

[10]We did not include GPTQ-LoRA in instruction-tuning experiments because the training was unstable.

[11]However we do not use an ELO-style rating system (which would require evaluations across all possible pairs) due to the large number of models involved.

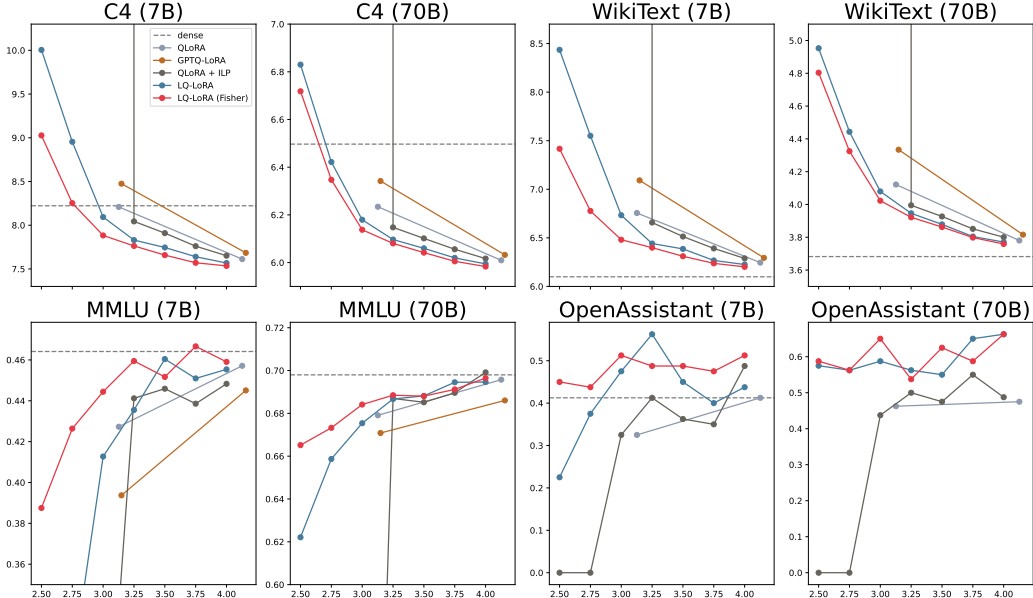

**Figure 2:** LQ-LoRA LLaMA-2 models with rank = 64. C4/Wikipedia/MMLU results are based on finetuning on C4. Vicuna eval is based on finetuning on the OpenAssistant dataset. QLoRA (Dettmers et al., 2023a) and GPTQ-LoRA (Chai et al., 2023) are based on our own reimplementations. Dense refers to unquantized models (no training) except for instruction tuning experiments. In the latter case, dense refers to full finetuning (7B model only, OOM for 70B).

**Training details.** Unless specified otherwise, we use a rank of 64, no LoRA dropout, and a default learning rate of $2 \times 10^{-5}$, with a few exceptions. For continual language modeling, we train on one partition of the C4 data for half an epoch, using a sequence length of 1024 for both training and evaluation. To estimate the Fisher, we use 10000 samples from C4 with a sequence length of 1024. For the GLUE tasks, we use a similar setup, but with masked language modeling objectives on C4. For instruction tuning, we use the hyperparameters suggested by Dettmers et al. (2023a) (except LoRA dropout). For GLUE fine-tuning, we follow the learning rate and number of epochs recommended by Hu et al. (2022) for the QLoRA baseline. However, we only fine-tune the model for 5 epochs for MNLI and QQP due to their sizes.

### 4.1 RESULTS

Figure 2 shows the results of language modeling and instruction tuning on LLaMA-2 across different model sizes and metrics. The full numeric results in Table 6 of Appendix C. In general we find that LQ-LoRA almost always outperforms QLoRA and GPTQ-LoRA at (near) similar bit budgets. For example, 3.5 bit (Fisher) LQ-LoRA is generally comparable to NF-4-bit QLoRA (which requires 4.127 bits/param); similarly, 2.75-bit LQ-LoRA is competitive with NF-3-bit QLoRA (which requires 3.127 bits/param). These comparisons highlight the utility of the mixed-quantization scheme since these mixed strategies would not even have been found without the ILP. It should be noted, however, that as we approach the 2.5-bit range, performance begins to degrade significantly. At the smaller 7B scale, the Fisher-weighted version of LQ-LoRA outperforms the unweighted version by a significant margin at all target bit widths. However, this discrepancy shrinks at the 70B scale.

| Method | Bits | GLUE |
|---|---|---|
| Full FT | 16 | 88.5 |
| QLoRA 3-bit | 3.127 | 86.1 |
| QLoRA | 2.5 | 75.4 |
| (ILP) | 2.75 | 80.7 |
| | 3.0 | 85.5 |
| | 3.25 | 86.1 |
| LQ-LoRA | 2.5 | 85.7 |
| | 2.75 | 87.1 |
| | 3.0 | 87.3 |
| | 3.25 | 88.1 |
| LQ-LoRA | 2.5 | 87.3 |
| (Fisher) | 2.75 | 86.4 |
| | 3.0 | 87.3 |
| | 3.25 | 88.3 |

**Table 1:** Performance on GLUE with RoBERTa-Large.

Table 1 shows GLUE benchmark w/ RoBERTa-Large, where we observe similar trends: LQ-LoRA outperforms QLoRA at similar bit-widths, and Fisher-weighted LQ-LoRA is effective at 2.5 bits.

### 4.2 LQ-LoRA FOR MODEL COMPRESSION

**Results.** Table 2 shows the results on C4 and WikiText, where we follow prior PTQ works (Frantar et al., 2022; Shao et al., 2023) and measure performance through C4 and WikiText-2 perplexity on a specific subset of data. LQ-LoRA with 2.75 bits results in an average bits/param of 2.95 bits and 2.85 bits for the 7B and 70B models respectively, when taking into account the LoRA components

**Table 2:** LQ-LoRA comparison against other sub-4-bit PTQ methods. While we only experiment with LQ-LoRA on LLaMA-2 (bottom), we show other PTQ results on LLaMA-1 (Table 7) as well to calibrate our results, as most prior works have focused on LLaMA-1. "Effective bits" takes into account the extra storage needed to store quantization parameters (e.g., scaling factors). In LQ-LoRA this includes the LoRA components, which are themselves quantized to 8 bits. For other methods, we take results corresponding to a setting with 3-bit quantization and a group-size 128 (if possible, otherwise the closest one). The effective bits for LQ-LoRA are dependent on model size, hence we show the effective bits for both settings. †Results from Shao et al. (2023).

| Method | Effective Bits (7B, 65B/70B) | C4 7B | C4 65B/70B | WikiText 7B | WikiText 65B/70B |
|---|---|---|---|---|---|
| *LLaMA-2 Uncompressed†* | *16* | *6.97* | *5.52* | *5.47* | *3.31* |
| RTN (3-bits, g128)† | 3.15 | 8.40 | 6.02 | 6.66 | 3.97 |
| GPTQ (3-bits, g128) (Frantar et al., 2022)† | 3.15 | 7.89 | **5.85** | 6.29 | 3.85 |
| AWQ (3-bits, g128) (Lin et al., 2023)† | 3.15 | 7.84 | - | 6.24 | - |
| OmniQuant (3-bits, g128) (Shao et al., 2023)† | 3.15 | 7.75 | **5.85** | 6.03 | 3.78 |
| OmniQuant (2-bits, g64) (Shao et al., 2023)† | 2.28 | 12.72 | 7.88 | 9.62 | 6.11 |
| LQ-LoRA (2.75-bits, 64-rank, Fisher) | 2.95, 2.85 | **7.60** | 5.88 | **5.67** | **3.65** |

**Table 3:** Performance on HuggingFace's Open LLM benchmark with LLaMA-2. We use the same LQ-LoRA setup as in Table 2 (2.75-bits, 64-rank, Fisher).

| Method | Size | ARC | HellaSwag | MMLU | TruthfulQA | Winogrande | GSM8K | Average |
|---|---|---|---|---|---|---|---|---|
| Uncompressed (16 bits) | 7B | 53.2 | 78.6 | 39.0 | 46.6 | 73.6 | 14.9 | 51.0 |
| LQ-LoRA (2.95 bits) | 7B | 49.8 | 75.9 | 39.3 | 43.0 | 72.4 | 7.4 | 48.0 |
| Uncompressed (16 bits) | 70B | 67.2 | 87.3 | 44.8 | 69.6 | 83.7 | 53.7 | 67.7 |
| LQ-LoRA (2.85 bits) | 70B | 65.8 | 86.2 | 44.5 | 66.9 | 83.2 | 45.6 | 65.3 |

("Effective bits" in Table 2). We find that this generally outperforms other sub-4-bit PTQ methods which also use calibration data to quantize the pretrained models.

Given the promising results with LQ-LoRA on continual language modeling, we next experiment with whether larger-scale language modeling can improve results further and enable the use of LQ-LoRA as a viable technique for model compression. Specifically, we take LQ-LoRA (Fisher, 2.75-bits, 64-rank) and fine-tune it on a larger calibration dataset of two C4 partitions and WikiText-2, using a sequence length of 2048. We further quantize the low-rank components themselves using NF-8 configuration after training.[12]

In Table 3, we evaluate the zero/few-shot capabilities using the Eleuther AI Language Model Evaluation Harness (Gao et al., 2023), a unified framework to test generative language models on a large number of different evaluation tasks. Specifically, we follow HuggingFace's the Open LLM Leaderboard[13] and evaluate models on 6 key benchmarks: ARC (Clark et al., 2018), HellaSwag (Zellers et al., 2019), MMLU (Hendrycks et al., 2020), TruthfulQA (Lin et al., 2022), Winogrande (Sakaguchi et al., 2021), and GSM8k (Cobbe et al., 2021). We observe that there is nontrivial degradation on some benchmarks (GSM8K, ARC), indicating that perplexity degradations are not always commensurate with downstream zero/few-shot perform

### 4.3 Analysis

**Mixed-configuration quantization.** We show the allocations of quantization configuration, measured by the average bits per parameter for a given matrix, in Figure 5. Each plot displays the decisions of the ILP for 2.75 target bit rate. ILP is able to allocate different configurations to different matrices, and this decision is indeed different between Fisher-weighted and non-Fisher-weighted variants.

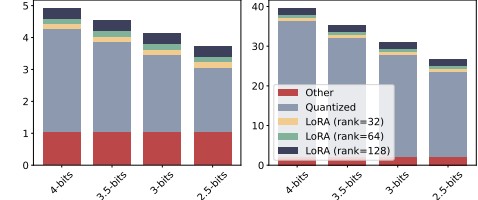

**Figure 3:** Storage (GB) broken down by quantized, LoRA, and other parameters. LLaMA-2 7B and 70B with 16-bits requires 14GB and 139GB.

**LoRA ranks.** We investigate the effect of LoRA rank with the LLaMA-2-7b model in Table 4 fixing the quantization configuration to NF-3 (i.e., 3.127 bits/param). QLoRA is insensitive to the LoRA rank. However, LQ-LoRA is able to make "better" use of the additional ranks to minimize the error at initialization, leading to improved performance.

**Memory requirements.** In Figure 3 we show the memory required for storing the model for different bit rates, broken down by the non-quantized component, the quantized component, and the

---

[12]NF-8 replaces the first-level quantization of the original NF-4 with 8-bits.

[13]https://huggingface.co/spaces/HuggingFaceH4/open_llm_leaderboard

**Table 4:** C4 and WikiText perplexity as a function of LoRA ranks. For both QLoRA and LQ-LoRA we used fixed NF-3 configuration for all layers, which incurs an average cost of 3.127 bits/param. The rightmost column shows the sum of errors across all matrices in LLaMA-2-7b, which corresponds to $\|\mathbf{W} - \text{Quantize}(\mathbf{W})\|_F^2$ for QLoRA and $\|\mathbf{W} - (\mathbf{Q} + \mathbf{L}_1\mathbf{L}_2)\|_F^2$ for LQ-LoRA.

| Method | LoRA rank | C4 | WikiText | Error |
|---|---|---|---|---|
| QLoRA 3-bit | 32 | 8.21 | 6.75 | |
| (3.127 bits/param) | 64 | 8.21 | 6.76 | $9.83 \times 10^4$ |
| | 128 | 8.21 | 6.76 | |
| LQ-LoRA 3-bit | 32 | 8.02 | 6.61 | $7.99 \times 10^4$ |
| (3.127 bits/param) | 64 | 7.93 | 6.51 | $7.12 \times 10^4$ |
| | 128 | 7.84 | 6.46 | $5.98 \times 10^4$ |

LoRA parameters. Quantization into sub-3-bits greatly decreases the memory required for running the model. At sub-3 bits, it becomes possible to run the 70B model on a single GPU with 40GBs. Finetuning requires more memory due to memory required for the activations and LoRA gradients/optimizer states. However, we are able to run full forward/backward passes on the sub-3-bit 70B models on a single 80GB GPU with batch size 2 and sequence length 2048.

## 5 DISCUSSION AND LIMITATIONS

Our simple iterative algorithm was found to be empirically effective but is ultimately heuristic, and it would be interesting to see if more theoretically-principled optimization algorithms could be derived. And while we focused on NF-quantization to enable comparison against QLoRA, applying LQ decomposition on top of other quantization approaches could result in further gains. It is also possible to extend the ILP-based mixed-precision approach to mixed-precision quantization *and* mixed-rank decomposition to enable the assignment of different ranks to each matrix.

We also discuss some negative results as well as limitations of LQ-LoRA. We found that re-factorizing the matrix periodically (e.g., after every $K$ gradient steps) did not yield improvements. Insofar as our initialization of $\mathbf{L}_1$ and $\mathbf{L}_2$ could orient the model to adapt itself in ways that may not be optimal for the task at hand, we also tried a hybrid approach where half of the adaptable low-rank component comes from LQ-LoRA, and the other half comes from standard LoRA initialization, but did not find this to improve results. Our approach also heavily relies on the fact that adaptation will occur through low-rank updates, and thus is not generally applicable to other parameter-efficient finetuning methods.

## 6 RELATED WORK

**Parameter-efficient finetuning.** Our work is closely related to parameter-efficient finetuning, such as Adapters (Houlsby et al., 2019; Mahabadi et al., 2021), prompt tuning (Li & Liang, 2021; Lester et al., 2021), and other methods which update subparts of the parameter vector (Guo et al., 2021; Zaken et al., 2022; Sung et al., 2021; Hu et al., 2022). Recent work also combines parameter-efficient finetuning methods with quantization (Kwon et al., 2022; Dettmers et al., 2023a).

**Low-rank plus sparse/quantized matrix decomposition.** Decomposing a data matrix into a low-rank matrix plus a sparse matrix (also known as robust PCA) is well-studied from both theoretical and applied perspectives (Lin et al., 2010; Zhou & Tao, 2011; 2013; Liu et al., 2013; Aravkin et al., 2014; Hintermuller & Wu, 2014; Yi et al., 2016; Zhang & Yang, 2017, *inter alia*). Within deep learning robust PCA has previously been applied to compress smaller models with fewer than 100M parameters (Chen & Ranftl, 2018; Cai et al., 2021). Saha et al. (2023) uses sketching techniques to obtain a quantized, low-rank approximation of a pretrained matrix. Recent contemporaneous work (Li et al., 2023) also performs low-rank plus quantized decomposition for LLM adaptation.

**LLM compression.** While there has been much work on low-rank compression of smaller LLMs with fewer than 1B parameters (Chen et al., 2021; Tukan et al., 2021; Tahaei et al., 2021), low-rank approaches for 1B+ LLMs remain underexplored, possibly because singular values of the pretrained matrices of LLMs have been found to decay slowly (Chen et al., 2021). Existing approaches for LLM compression have thus generally focused on quantization. Much recent work has focused on data-aware quantization strategies (Dettmers et al., 2022; Xiao et al., 2022; Dettmers et al., 2023b; Frantar et al., 2022; Kim et al., 2023b; Lin et al., 2023).

## 7 CONCLUSION

This work proposes a simple extension of LoRA which factorizes the pretrained matrices into low-rank and quantized components, where the quantization component can employ a dynamic configuration strategy. We observed this low-rank plus quantized decomposition approach to yield meaningful improvements over strong baselines.

ACKNOWLEDGEMENTS

We thank Minyoung Huh, Isha Puri, Hongyi Wang, Lirui Wang, and the members of the Hyundai 42dot research team for helpful comments and discussions. We are also grateful to Mengzhao Chen for clarification questions regarding OmniQuant. Eric Xing and Han Guo acknowledge the support of Microsoft PhD Fellowship, NGA HM04762010002, NSF IIS1955532, NIGMS R01GM140467, NSF IIS2123952, NSF BCS2040381, NSF IIS2311990, SRC AIHW 2024AH3210, and DARPA ECOLE HR00112390063. This study was supported by funds from Hyundai Motor Group, MIT-IBM Watson AI, and the MLA@CSAIL initiative.

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

## A    DATA-AWARE MATRIX DECOMPOSITION VIA FISHER-WEIGHTED SVD

The decomposition objective considered in §3.1 is data-agnostic insofar as it treats each entry of $\mathbf{W}$ as equally important for reconstruction during factorization. Following recent works which demonstrate the importance of using calibration data for quantizating LLMs (Frantar et al., 2022; Lin et al., 2023; Kim et al., 2023b), we next consider a data-aware version of the approach by using a diagonal approximation of the Fisher information matrix to weight the reconstruction objective. The (diagonal of the) empirical Fisher information matrix for $\mathbf{W}$ is given by $\mathbf{F} \in \mathbb{R}^{d \times k}$ where each entry of the matrix is the averaged square of the derivative over $D$ samples, i.e., $\mathbf{F}_{ij} = \frac{1}{D} \sum_{d=1}^{D} \left( \frac{\partial}{\partial \mathbf{W}_{ij}} \log p_{\text{LM}} \left( \mathbf{x}^{(d)} \right) \right)^2$. Intuitively, this metric measures how sensitive the model's output is to a perturbation of each parameter, and has previously been exploited to improve low-rank compression (Hsu et al., 2022) and quantization (Kim et al., 2023b) of pretrained language models. We similarly use $\mathbf{F}$ to weight the decomposition objective,

$$\left\| \sqrt{\mathbf{F}} \odot \left( \mathbf{W} - \left( \mathbf{Q} + \mathbf{L}_1 \mathbf{L}_2 \right) \right) \right\|_F^2, \tag{5}$$

where $\odot$ is the Hadamard product. When applied to the LQ decomposition algorithm from §3.1, this results in the following weighted SVD problem, where given $\mathbf{E} := \mathbf{W} - \mathbf{Q}$ and weighting matrix $\mathbf{F}$, we must find matrices $\mathbf{L}_1 \in \mathbb{R}^{d \times r}, \mathbf{L}_2 \in \mathbb{R}^{r \times k}$ that form the best rank-$r$ approximation,

$$\mathbf{L}_1, \mathbf{L}_2 = \underset{\mathbf{L}_1, \mathbf{L}_2}{\arg\min} \, \left\| \sqrt{\mathbf{F}} \odot \left( \mathbf{E} - \mathbf{L}_1 \mathbf{L}_2 \right) \right\|_F^2.$$

Unliked its unweighted counterpart, this problem is in general intractable (and in fact NP-hard; Razenshteyn et al., 2016) and is typically addressed through approximate methods (Srebro & Jaakkola, 2003; Li et al., 2016; Tuzhilina & Hastie, 2021). However, if we assume that either rows or columns of the weight matrix $\mathbf{F}$ have identical values, we have the following identity,

$$\mathbf{L}_1, \mathbf{L}_2 = \underset{\mathbf{L}_1, \mathbf{L}_2}{\arg\min} \, \left\| \sqrt{\mathbf{F}} \odot \left( \mathbf{E} - \mathbf{L}_1 \mathbf{L}_2 \right) \right\|_F^2 \quad = \underset{\mathbf{L}_1, \mathbf{L}_2}{\arg\min} \, \left\| \mathbf{D}_{\text{row}} \left( \mathbf{E} - \mathbf{L}_1 \mathbf{L}_2 \right) \mathbf{D}_{\text{col}} \right\|_F^2,$$

where $\mathbf{D}_{\text{row}}$ is a diagonal matrix consists of row-means of $\sqrt{\mathbf{F}}$, and $\mathbf{D}_{\text{col}}$ is a diagonal matrix consisting of the column-means of $\sqrt{\mathbf{F}}$, i.e.,

$$\mathbf{D}_{\text{row}} = \text{diag} \left( \left[ \text{avg}(\sqrt{\mathbf{F}_{1,\cdot}}), \ldots, \text{avg}(\sqrt{\mathbf{F}_{d,\cdot}}) \right] \right), \mathbf{D}_{\text{col}} = \text{diag} \left( \left[ \text{avg}(\sqrt{\mathbf{F}_{\cdot,1}}), \ldots, \text{avg}(\sqrt{\mathbf{F}_{\cdot,k}}) \right] \right).$$

In this case the above problem can be solved exactly by standard SVD,

$$\mathbf{U}, \mathbf{\Sigma}, \mathbf{V}^\top \leftarrow \text{SVD}(\mathbf{D}_{\text{row}} \mathbf{A} \mathbf{D}_{\text{col}}), \quad \mathbf{L}_1 \leftarrow \mathbf{D}_{\text{row}}^{-1} \mathbf{U} \sqrt{\mathbf{\Sigma}}, \qquad \mathbf{L}_2 \leftarrow \sqrt{\mathbf{\Sigma}} \mathbf{V}^\top \mathbf{D}_{\text{col}}^{-1}. \tag{6}$$

(See Algorithm 4.) While the homogenous row/column assumption clearly does not hold for $\mathbf{F}$, we found this approach to work well in practice.[14] We note that this approximation is a simple extension of Hsu et al. (2022) who use $\mathbf{D}_{\text{row}}$ but not $\mathbf{D}_{\text{col}}$ in their weighted SVD (we found that using both the row- and column-averages performed slightly better).

**Discussion.** This data-aware version of LQ-LoRA requires being able to backpropagate through the pretrained LM in order to obtain the Fisher matrices $\{\mathbf{F}^{(i)}\}_{i \in [N]}$, which, in some sense, goes against the setting targeted by memory-efficient adaptation methods wherein full finetuning is not considered possible. This is a valid point, and hence we study both version of LQ-LoRA in our empirical study. We note however, that we compute $\{\mathbf{F}^{(i)}\}_{i \in [N]}$ based on some generic text data to obtain the LQ-LoRA initializations $\{\mathbf{Q}^{(i)}, \mathbf{L}_1^{(i)}, \mathbf{L}_2^{(i)}\}_{i \in [N]}$, and use the *same* initialization for different downstream tasks. This makes the data-aware approach practical, as the Fisher computaton and the matrix decomposition needs to performed only once (as in the non-data-aware version).

## B    IMPLEMENTATION DETAILS

Here we discuss some implementation details for efficiently implementing LQ-LoRA.

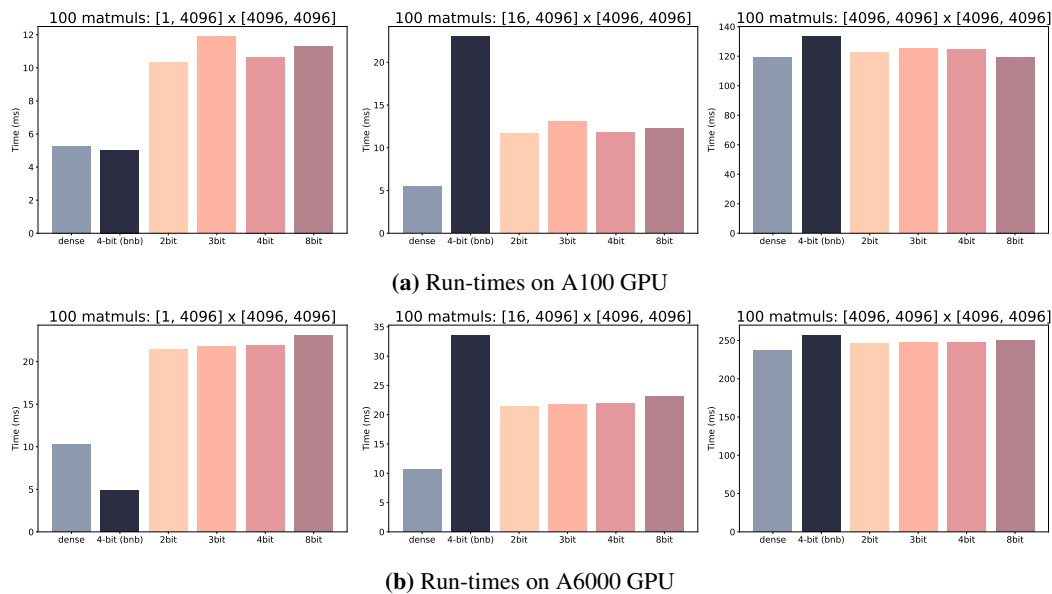

**(a)** Run-times on A100 GPU

**(b)** Run-times on A6000 GPU

**Figure 4:** A100/A6000 GPU runtime to perform 100 matrix-matrix multiplications in `fp32` between input data and quantized matrices (which involves on the fly dequantization). `bitsandbytes` (bnb) (Dettmers et al., 2023a) has separate implementations for training and for inference (matrix-vector multiplications, leftmost figure). We use the same quantization configuration as NF-4 and vary the first-level bits $(2, 3, 4, 8)$ for consistent comparisons.

**Search Space.** Please see Table 5.

**PyTorch-based mixed-quantization.** Weight-only quantization techniques typically require packing sub-8-bit matrices into natively-supported data types (e.g., `int8`), and then unpacking to float-point format during dequantization. As such, existing implementations often require custom CUDA extensions that are dependent on a particular quantization configuration, making it difficult to extend to mixed-quantization strategies. Our implementation is based entirely on PyTorch for fast experimentation and implementation of dynamic quantization strategies. We use PyTorch's `__torch_dispatch__` functionality to duck-type `torch.Tensor`,[15] which redefines behaviors

| Configuration grid | |
|---|---|
| $b_0$ | $\{2, 3, 4\}$ |
| $b_1$ | $\{2, 3, 4\}$ |
| $b_2$ | $\{\text{bf16}, \text{fp16}, \text{fp32}\}$ |
| $B_0$ | $\{16, 32, 64\}$ |
| $B_1$ | $\{16, 64, 256\}$ |

**Table 5:** Search space $\mathcal{C}$ for the NF quantization configurations.

under PyTorch operations such as addition and matrix multiplication. We then use PyTorch's (full-graph) compiler to compile the following operations: (1) bits-unpacking, (2) dequantization, (3) linear algebra operations such as `add` and `matmul`, and (4) transpose and casting (for `bf16` training). For LoRA finetuning, we observed this PyTorch-based implementation (followed by compilation) to be as fast as QLoRA's `bitsandbytes` implementation,[16] which relies heavily on CUDA extensions that are tailored for 4-bit NF quantization.

**LoRA optimizer offloading.** We also optionally work with a CPU-based optimizer (Ren et al., 2021), which extends the pageable optimizer proposed in Dettmers et al. (2023a). This implementation takes advantage of the fact that in LoRA, only a small portion of parameters needs to be trained, which makes data movement between CPU and GPU, as well as computation on CPU, relatively

---

[14]In preliminary experiments we also explored a version of data-aware LQ-LoRA where we approximately minimized $\|\mathbf{X}(\mathbf{W} - (\mathbf{Q} + \mathbf{L}_1\mathbf{L}_2))\|_F$ using activations $\mathbf{X}$ from calibration data, instead of the Fisher information matrix. However we found this to underperform the Fisher approach.

[15]https://dev-discuss.pytorch.org/t/what-and-why-is-torch-dispatch/557

[16]https://github.com/TimDettmers/bitsandbytes

**Table 6:** LQ-LoRA LLaMA-2 models with rank = 64. C4/Wikipedia/MMLU results are based on finetuning on C4. Vicuna eval is based on finetuning on the OpenAssistant dataset. QLoRA (Dettmers et al., 2023a) and GPTQ-LoRA are based on our own reimplementations.

| Method | Bits per param | C4 (PPL) | | WikiText (PPL) | | MMLU (acc.) | | Vicuna Eval | |
|---|---|---|---|---|---|---|---|---|---|
| | | 70B | 7B | 70B | 7B | 70B | 7B | 70B | 7B |
| Dense (no training) | - | 6.50 | 8.22 | 3.68 | 6.10 | 0.70 | 0.46 | - | - |
| Dense (full finetuning) | - | - | - | - | - | - | - | OOM | 0.41 |
| QLoRA 3-bit | 3.127 | 6.23 | 8.21 | 4.12 | 6.76 | 0.68 | 0.43 | 0.46 | 0.33 |
| QLoRA 4-bit | 4.127 | 6.01 | 7.61 | 3.78 | 6.25 | 0.70 | 0.46 | 0.47 | 0.41 |
| GPTQ-LoRA 3-bit | 3.148 | 6.34 | 8.48 | 4.33 | 7.09 | 0.67 | 0.39 | - | - |
| GPTQ-LoRA 4-bit | 4.156 | 6.03 | 7.68 | 3.82 | 6.29 | 0.69 | 0.45 | - | - |
| QLoRA + ILP | 2.50 | 2223.2 | 2996.3 | 3319.4 | 4084.3 | 0.23 | 0.23 | 0.00 | 0.00 |
| | 2.75 | 2193.9 | 2736.5 | 3292.6 | 3932.2 | 0.23 | 0.27 | 0.00 | 0.00 |
| | 3.00 | 1781.5 | 1969.3 | 2587.0 | 3091.0 | 0.23 | 0.23 | 0.44 | 0.33 |
| | 3.25 | 6.15 | 8.04 | 3.99 | 6.66 | 0.69 | 0.44 | 0.50 | 0.41 |
| | 3.50 | 6.10 | 7.91 | 3.93 | 6.51 | 0.69 | 0.45 | 0.47 | 0.36 |
| | 3.75 | 6.06 | 7.76 | 3.85 | 6.39 | 0.69 | 0.44 | 0.55 | 0.35 |
| | 4.00 | 6.02 | 7.65 | 3.80 | 6.29 | 0.70 | 0.45 | 0.49 | 0.49 |
| LQ-LoRA | 2.50 | 6.83 | 10.00 | 4.95 | 8.44 | 0.62 | 0.31 | 0.57 | 0.23 |
| | 2.75 | 6.42 | 8.95 | 4.44 | 7.55 | 0.66 | 0.31 | 0.56 | 0.38 |
| | 3.00 | 6.18 | 8.09 | 4.08 | 6.73 | 0.68 | 0.41 | 0.59 | 0.47 |
| | 3.25 | 6.10 | 7.83 | 3.95 | 6.44 | 0.69 | 0.44 | 0.56 | 0.56 |
| | 3.50 | 6.06 | 7.75 | 3.88 | 6.39 | 0.69 | 0.46 | 0.55 | 0.45 |
| | 3.75 | 6.02 | 7.64 | 3.80 | 6.27 | 0.69 | 0.45 | 0.65 | 0.40 |
| | 4.00 | 5.99 | 7.57 | 3.77 | 6.23 | 0.69 | 0.46 | 0.66 | 0.44 |
| LQ-LoRA (Fisher) | 2.50 | 6.72 | 9.03 | 4.80 | 7.42 | 0.67 | 0.39 | 0.59 | 0.45 |
| | 2.75 | 6.35 | 8.25 | 4.32 | 6.78 | 0.67 | 0.43 | 0.56 | 0.44 |
| | 3.00 | 6.14 | 7.88 | 4.02 | 6.48 | 0.68 | 0.44 | 0.65 | 0.51 |
| | 3.25 | 6.08 | 7.76 | 3.92 | 6.40 | 0.69 | 0.46 | 0.54 | 0.49 |
| | 3.50 | 6.04 | 7.66 | 3.86 | 6.31 | 0.69 | 0.45 | 0.62 | 0.49 |
| | 3.75 | 6.01 | 7.57 | 3.80 | 6.24 | 0.69 | 0.47 | 0.59 | 0.47 |
| | 4.00 | 5.98 | 7.53 | 3.76 | 6.20 | 0.70 | 0.46 | 0.66 | 0.51 |

manageable. We retain a copy of trainable parameters on CPU, offload gradients from GPU to CPU before executing optimizer step on the parameter copy on CPU, and copy them back into GPU. We overlap the per-matrix optimizer step and CPU to GPU movement through async copy. On the largest 70 billion parameter model, we noticed a 14% memory saving with only a marginal (<2%) increase in training speed with this strategy.

**Runtimes.** Figure 4 displays the run-times of matrix multiplications between FP32 input data and quantized matrices. We dequantize the matrix just-in-time before executing the matmul, and hence the runtime is lower-bounded by FP32 matmul (dense). We enable TF32 in PyTorch to utilize Tensor Cores, and collect CUDA time through PyTorch Profiler. Notably, the dequantization overhead is relatively small for reasonably-sized matrices.

## C   FULL RESULTS

Table 6 shows the full numeric results of LQ-LoRA vs. QLoRA and GPT-LoRA. Table 7 shows extended PTQ results including those on LLaMA-1 (top) from prior works. Figure 5 shows the allocations of quantization configuration, measured by the average bits per parameter for a given matrix.

**Table 7:** LQ-LoRA comparison against other sub-4-bit PTQ methods. While we only experiment with LQ-LoRA on LLaMA-2 (bottom), we show other PTQ results on LLaMA-1 (top) as well to calibrate our results, as most prior works have focused on LLaMA-1. "Effective bits" takes into account the extra storage needed to store quantization parameters (e.g., scaling factors). In LQ-LoRA this includes the LoRA components, which are themselves quantized to 8 bits. For other methods, we take results corresponding to a setting with 3-bit quantization and a group-size 128 (if possible, otherwise the closest one). The effective bits for SpQR and LQ-LoRA are dependent on model size, and hence we show the effective bits for both settings. [†]Results from Shao et al. (2023). [‡]We show 3.1-bits instead of 3.01-bits with group size 128 because the latter performed worse.

| Method | Effective Bits (7B, 65B/70B) | C4 | | WikiText | |
| --- | --- | --- | --- | --- | --- |
| | | 7B | 65B/70B | 7B | 65B/70B |
| *LLaMA-1 Uncompressed*[†] | *16* | *7.08* | *5.62* | *5.68* | *3.53* |
| SpQR (Dettmers et al., 2023b) | 3.94, 3.90 | 7.28 | 5.70 | 5.87 | 3.68 |
| RTN (3-bits, g128)[†] | 3.15 | 8.62 | 6.10 | 7.01 | 4.24 |
| GPTQ (3-bits, g128) (Frantar et al., 2022)[†] | 3.15 | 7.85 | 6.00 | 6.55 | 4.17 |
| AWQ (3-bits, g128) (Lin et al., 2023)[†] | 3.15 | 7.92 | 5.94 | 6.46 | 3.99 |
| PEQA (3-bits, g128) (Kim et al., 2023a) | 3.15 | - | - | 5.91 | - |
| OWQ (3-bits) (Lee et al., 2023)[‡] | 3.1 | 8.15 | 6.16 | 6.39 | 4.08 |
| SqueezeLLM (3-bits, 0.45%) (Kim et al., 2023b) | 3.24 | 7.56 | - | 6.13 | - |
| SqueezeLLM (3-bits) (Kim et al., 2023b) | 3.02 | 7.75 | - | 6.32 | - |
| OmniQuant (3-bits, g128) (Shao et al., 2023)[†] | 3.15 | 7.75 | 5.93 | 6.15 | 3.94 |
| OmniQuant (2-bits, g64) (Shao et al., 2023)[†] | 2.28 | 11.78 | 7.60 | 8.90 | 5.65 |
| LREC (3-bits, g128) (Chai et al., 2023) | 3.35 | 8.24 | - | 5.52 | - |
| LREC (2-bits, g128) (Chai et al., 2023) | 2.24 | 12.52 | - | 8.74 | - |
| *LLaMA-2 Uncompressed*[†] | *16* | *6.97* | *5.52* | *5.47* | *3.31* |
| RTN (3-bits, g128)[†] | 3.15 | 8.40 | 6.02 | 6.66 | 3.97 |
| GPTQ (3-bits, g128) (Frantar et al., 2022)[†] | 3.15 | 7.89 | **5.85** | 6.29 | 3.85 |
| AWQ (3-bits, g128) (Lin et al., 2023)[†] | 3.15 | 7.84 | - | 6.24 | - |
| OmniQuant (3-bits, g128) (Shao et al., 2023)[†] | 3.15 | 7.75 | **5.85** | 6.03 | 3.78 |
| OmniQuant (2-bits, g64) (Shao et al., 2023)[†] | 2.28 | 12.72 | 7.88 | 9.62 | 6.11 |
| LQ-LoRA (2.75-bits, 64-rank, Fisher) | 2.95, 2.85 | **7.60** | 5.88 | **5.67** | **3.65** |

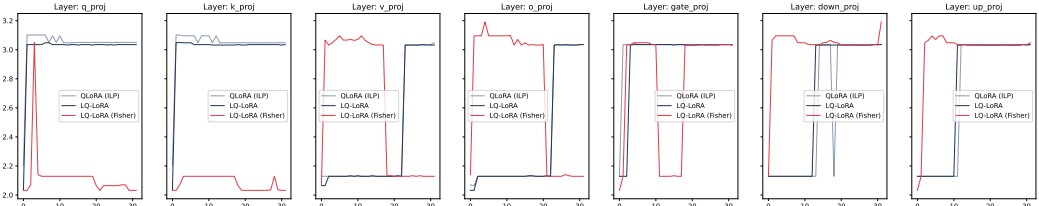

**Figure 5:** Visualization of the bits/param allocated by ILP broken down by matrix type. The y-axis is the bits/param, while x-axis indicates layer number. We show the allocation for QLoRA, LQ-LoRA and Fisher-weighted LQ-LoRA for target bit rate of 2.75 bits.

