# OpenReview forum: "LQ-LoRA: Low-rank plus Quantized Matrix Decomposition for Efficient Language Model Finetuning"
_ICLR.cc/2024/Conference — ICLR 2024 poster_

### Official Review · Reviewer_4gD2 · 2023-10-30

**Soundness:** 2 fair
**Presentation:** 3 good
**Contribution:** 2 fair
**Rating:** 5
**Confidence:** 4

**Summary:**

The authors of the paper propose an iterative method to decompose a pre-trained weight matrix W into a quantized component $Q$ and a low-rank component $L_1L_2$, which encourages $Q + L_1L_2$ to approximate $W$ as much as possible. They also present dynamic configuration quantization via integer linear programming and a data-aware decomposition method by employing the Fisher information matrix.

**Strengths:**

The authors of the paper the first time tackles the issue that $Q + L_1L_2$ might not be equal to its original pre-trained weight matrix $W$. Moreover, they try to assign different configurations to each pre-trained weight matrix by utilizing integer linear programming.

**Weaknesses:**

It seems necessary that LQ-LoRA should be compared to the case when BRECQ, OPTQ, and/or FlexRound is used to quantize pre-trained weight matrices with $L_2$ initialized to zero. The reason why I bring this up is that if BRECQ, OPTQ, and/or FlexRound is employed for $q(\cdot)$, $X(q(W) + L_1L_2)$ would be approximately equal to $XW$ because all their objectives are designed to minimize the difference between $ Xq(W)$ and $XW$. Then, the need for the proposed data-aware decomposition method would be marginal. Although the authors mention that they stick to NF quantization, integer quantization can be also surely used in QLoRA.

**Questions:**

None

---

> ### Author Response · Authors · 2023-11-16
> **Response**
>
> Thanks for the comments --- please see our responses below.
>
> **Comparison with methods that use the data $X$ to quantize**
>
> Thanks for the question! We actually did implement and compare against a baseline that uses OPTQ/GPTQ to quantize first before learning low-rank updates. In particular, we use GPTQ to first quantize the model and then learn low-rank updates. We called this "GPTQ-LoRA" baseline "LREC" in the paper following this paper (https://arxiv.org/pdf/2306.08162.pdf). We find that GPTQ-LoRA underperforms both QLoRA and LQ-LoRA (Table 1).
>
> (We now realize that this naming caused some confusion. Thanks for pointing that out --- we will clarify this in the paper.)
>
>
> **Additional Post Training Quantization Experiments**
>
> In addition to the GPTQ-LoRA experiments already in the paper, we performed additional experiments using LQ-LoRA as a potential alternative to PTQ methods by training on a larger calibration dataset. Specifically, we train the LQ-LoRA (Fisher, 2.75-bit, 64-rank) for one epoch on two C4 partitions and WikiText-2. We then further quantize the LoRA components themselves to 8-bit (NF8). Please see the Table below for the results, where the numbers other than LQ-LoRA are from OmniQuant [1]. Note that we use the term "effective bits" to denote bits per parameter that treat the LoRA components as "overheads" for comparing with methods without LoRA; LoRA components (with NF-8 quantization) amount to about 0.1-0.2 extra bits per parameter.
>
>
>
> | Base Model | Method                           | Effective Bits (7B, 70B) | C4 (7B) | C4 (70B) | WikiText2 (7B) | WikiText2 (70B) |
> | -------- | ---------------------------------- | ----------- | ---- | ---- | ---- | ---- |
> | LLaMA-2 | Dense                               | 16          | 6.97 | 5.52 | 5.47 | 3.31 |
> | LLaMA-2 | RTN                                 | 3.15        | 8.40 | 6.02 | 6.66 | 3.97 |
> | LLaMA-2 | GPTQ            | 3.15        | 7.89 | 5.85 | 6.29 | 3.85 |
> | LLaMA-2 | AWQ               | 3.15        | 7.84 | -    | 6.24 | -    |
> | LLaMA-2 | OmniQuant  | 3.15        | 7.75 | 5.85 | 6.03 | 3.78 |
> | LLaMA-2 | LQ-LoRA (Fisher)   | 2.95, 2.85 | 7.60 | 5.88 | 5.67 | 3.65 |
>
>
>
>
> [1] https://arxiv.org/abs/2308.13137
>
> These results indicate that LQ-LoRA can also be used as a state-of-the-art PTQ method. Given the promise of LQ-LoRA as a potential alternative to PTQ, we will include this comparison against PTQ results in the next iteration of the paper.
>
>
>
> **NF-based vs INT-based Quantization**
>
> You are correct in that the techniques in this paper are not specific to NF. QLoRA paper demonstrated the superior quality of such a scheme compared to classical integer quantization. Our preliminary experiments confirmed this, and hence, we chose NF.
>
> We conducted the following experiment to be more concrete about NF vs INT. We ran LQ (i.e., Section 3.1) on LLaMA-2 7B's dense parameters with `rank=64` and NF3/INT3/NF4/INT4 quantization configuration (i.e., `b0=3/4, b1=8, b2=fp32, B0=64, B1=256`), and measure the reconstruction errors (i.e., $||\mathbf{W} - (\mathbf{Q} + \mathbf{L}_1 \mathbf{L}_2) ||_2^2$). The table below shows that NF indeed performed better.
>
> | Method      | Reconstruction error (unit: $10^4$) |
> | ----------- | ----------- |
> | LQ with NF3      | 7.12       |
> | LQ with INT3     | 11.6        |
> | LQ with NF4      | 1.42       |
> | LQ with INT4     | 2.02        |

---

> ### Author Response · Authors · 2023-11-21
> **Discussion**
>
> Hi there! Please let us know if the above response answered some of your questions in the original review, and please let us know if you have follow-up questions!
>
> In particular, we want to highlight a few key takeaways from our response:
> 1. We included a baseline that uses OPTQ/GPTQ to quantize first before learning low-rank updates. We called this "GPTQ-LoRA" baseline "LREC" in the paper.
> 2. We performed additional experiments using LQ-LoRA as a potential alternative to PTQ methods by training on a larger calibration dataset. (Please see the above response for more details.)
> 3. The techniques in this paper are not specific to NF. We chose NF because of its superior quality. To be more concrete about this, we conducted additional experiments. (Please see the above response for more details, too.)

---

> ### Comment · Reviewer_4gD2 · 2023-11-22
>
> Thank you for your detailed response.
>
> However, my concerns are not still completely addressed. First of all, the accuracy of LREC $4$-bit on MMLU almost matches that of LQ-LoRA, which seem to make the motivation of LQ-LoRA weaker. Furthermore, the authors compare their method with GPTQ-LoRA only. As GPTQ is a quantization method based on a layer-wise optimization, if a quantization method based on a block-wise optimization such as BRECQ [1] and FlexRound [2] is employed, "BRECQ-LoRA" or "FlexRound-LoRA" could surpass LQ-LoRA. Then, in my view, the motivation of LQ-LoRA would become very weak. To strengthen the motivation of LQ-LoRA, it would be necessary to compare LQ-LoRA with "BRECQ-LoRA" or "FlexRound-LoRA".
>
> [1] BRECQ: Pushing the Limit of Post-Training Quantization by Block Reconstruction, ICLR 2021
>
> [2] FlexRound: Learnable Rounding based on Element-wise Division for Post-Training Quantization, ICML 2023

---

> ### Author Response · Authors · 2023-11-22
> **Thanks for the response!**
>
> Thank you for the discussion! Please see our responses below.
>
> _1. The accuracy of LREC-4bit on MMLU almost matches that of LQ-LoRA_
>
> For both LLaMA-2 7B and 70B models, LQ-LoRA (Fisher) with 3.25 bits can match the performance of LREC/GPTQ-LoRA 4-bits (i.e., 4.156-bits). Similarly, LQ-LoRA (Fisher) with 2.5/2.75 bits can match the performance of LREC/GPTQ-LoRA 3-bits (i.e., 3.148-bits). Please see the table below.
>
> | Method | Bits per param | MMLU (70B) | MMLU (7B) |
> | ----- | ----- | ----- | ----- |
> | GPTQ-LoRA 3-bit |  3.148 | 0.67$^\dagger$ | 0.39 |
> | LQ-LoRA (Fisher) | 2.50 | 0.67 | 0.39 |
> | LQ-LoRA (Fisher) | 2.75 | 0.67 | 0.43 |
>
> | Method | Bits per param | MMLU (70B) | MMLU (7B) |
> | ----- | ----- | ----- | ----- |
> | LREC-4 bit | 4.156 | 0.69 | 0.45 |
> | LQ-LoRA (Fisher) | 3.25 | 0.69 | 0.46 |
>
> $^\dagger$Note that we additionally filled in the MMLU 70B performance for GPTQ-LoRA 3-bit; this number was absent from the initial submission. Based on the above, we conclude that LQ-LoRA can meaningfully improve upon GPTQ-LoRA and QLoRA.
>
> _2. The authors compare their method with GPTQ-LoRA only_
>
> Besides GPTQ-LoRA, we also compared our method QLoRA and different variants of LQ-LoRA. In addition, our latest PTQ experiments included comparisons with RTN, GPTQ, AWQ, and OmniQuant. For details, please see "Additional Post Training Quantization Experiments" in our earlier response.
>
> _3. BRECQ and FlexRound_
>
> Thanks for the suggestions! We think applying other quantization techniques is certainly interesting, but we would like to raise three points.
>
> a) For quantizing large language models with 10B+ parameters, our understanding is that  NF quantization (from QLoRA) and GPTQ --- methods we compared with --- are near state-of-the-art quantization methods. (While we show numbers from other methods, such as Omniquant, in the PTQ comparison table above, these are, as far as we know, preprints).
>
> b) Applying fancier quantization techniques such as BRECQ/Adaround to large models is challenging. For example, for BRECQ,  the OmniQuant paper [1] notes,
> > ... BRECQ ... cannot be applied in models with billions of parameters because they are hard to optimize due to the huge solution space.
>
> Similarly, the SpQR paper [2] stated that,
>
> > ... BRECQ ... were designed for vision models or small-scale language models, with less than 100M parameters.
>
> [1] https://arxiv.org/abs/2308.13137
>
> [2] https://arxiv.org/abs/2306.03078
>
> FlexRound is an interesting suggestion, but we were unable to find an open-source implementation of this. Hence, we focused our comparison on widely-used LLM quantization methods with open-source code, i.e., QLoRA and GPTQ-LoRA.
>
> c) Finally, (and most importantly), we note that LQ-LoRA can work with generic quantization approaches! For example, during each step of the iterative algorithm, instead of using NF-quantization, we can use other quantization methods to minimize $\Vert \mathbf{X}\mathbf{W} - \mathbf{X}(\operatorname{quantize}(\mathbf{W}) + \mathbf{L}_1 \mathbf{L}_2) \Vert$. We focused on NF quantization because the quantization function is extremely quick (on the order of seconds), and it performs on par with GPTQ despite being data-agnostic. But we think it's worth highlighting this aspect of LQ-LoRA more, and we will discuss this further.

---

> ### Comment · Reviewer_4gD2 · 2023-11-23
>
> The authors referenced comments including (1) and (2) as below.
>
> (1) ... BRECQ ... cannot be applied in models with billions of parameters because they are hard to optimize due to the huge solution space.
>
> (2) ... BRECQ ... were designed for vision models or small-scale language models, with less than 100M parameters.
>
> However, in the FlexRound paper, the experimental results of BRECQ are quite well for LLaMA 7B, 13B, and even 30B. In this sense, just referencing those comments seems not to be enough to reinforce the motivation of the paper. To confirm the validity of LQ-LoRA further, the paper needs to be more polished in my view.

---

### Official Review · Reviewer_CSBT · 2023-10-31

**Soundness:** 3 good
**Presentation:** 3 good
**Contribution:** 3 good
**Rating:** 6
**Confidence:** 4

**Summary:**

This paper proposes LQ-LoRA, a memory-efficient LLM adaptation method that decomposes each pretrained matrix into a high-precision low-rank component and a memory-efficient quantized component. The algorithm is adapted from QLoRA and applied modification to solve the problem that zero initialization of the low-rank matrix may not be optimal when the fixed matrix is quantized. The method decomposes the matrix by an iterative algorithm and updates only the low-rank matrix weights during fine-tuning. Results showed that the proposed method outperforms QLoRA and LREC with similar bit compression rates.

**Strengths:**

-	The proposed method decomposes the pretrained matrix into a quantizable fixed matrix and low-rank matrix that is already optimized before fine-tuning starts, which contributes to improved accuracy.
-	The paper shows that LQ-LoRA can be used as a mixed quantization strategy, and also proposes a data-aware version of the algorithm, which enables users to flexibly set a target memory budget.
-	Results show that the proposed method can be generalized to different model families by showing outperforming results with RoBERTa and LLaMA.

**Weaknesses:**

- The authors have introduced a method that employs an iterative algorithm for initialization. Can they provide insights regarding the computational latency associated with their approach?

- The authors assert the efficiency of LQ-LoRA based on empirical evidence, yet lack theoretical backing. To strengthen the credibility of the algorithm, a comparison might be beneficial, especially with methods that initialize the Q(W) + L1L2 matrix in a manner that closely mirrors the original pretrained matrix W. Consider, for instance, the use of GPTQ as a compensatory mechanism.

- It appears that this paper serves as an expanded or refined rendition of the Q-LoRA paper. As such, it seemingly inherits the same limitation, notably the inference overhead, given that this approach must fail to integrate the LoRA layer into an existing linear layer.

- Similarly, I would like to raise a query about the paper's novelty. While this method undeniably enhances the current approach (Q-LoRA), from a PEFT perspective, there could be superior methods, particularly concerning inference challenges. On the topic of novelty, I await the insights of fellow reviewers.

**Questions:**

Included in the weakness.

---

> ### Author Response · Authors · 2023-11-16
> **Response**
>
> Thanks to the reviewer for the comments! Please see our responses below.
>
> **Latency of The Iterative Algorithm**
>
> This is a one-time cost at the initialization and takes a fraction of the total training time. Each step of the iterative algorithm consists of SVD followed by quantization. Note that we use randomized SVD (rSVD) for speed consideration, and we run the algorithm for up to 100 steps (and we stop before this if the error starts to increase).
>
> To be more concrete, we measured the runtime to process three of the LLaMA-2 7B's matrices with `rank=64` and NF3 quantization. Please see the table below.
>
> | Data-Aware (i.e., Section 3.3) | Time / Matrix |
> | ----- | ----- |
> | No  | 1.3 to 2.1 seconds |
> | Yes | 1.3 to 2.3 seconds |
>
> **Comparison with GPTQ-based Methods**
>
> Thanks for the question! We actually did implement and compare against GPTQ-LoRA. We called this "LREC" in the paper following this paper (https://arxiv.org/pdf/2306.08162.pdf). We find that GPTQ-LoRA underperforms both QLoRA and LQ-LoRA (Table 1).
>
> (We now realize that this naming caused some confusion. Thanks for pointing that out --- we will clarify this in the paper.)
>
> **Limitations from QLoRA**
>
> This is a very good point! But we want to point out that inference with quantized + low-rank is not necessarily slower, especially in batch = 1 setting. Smaller quantized matrices allow us to reduce data movement between SRAM and HBM of the GPU. For example, in the latest releases of `bitsandbytes` (the low-level engine behind QLoRA) [1], they claimed their matrix-vector multiplication between quantized matrix and dense vector (the one needed during inference) could be competitive and even outperformed dense matrix-vector multiplication.
>
> [1] https://github.com/TimDettmers/bitsandbytes/releases
>
> *Integrating LoRA Layers Into Linear Layers*
>
> Given the potential inference speed-ups, we think that practitioners could use the "quantized + low-rank" models without merging. Such models could be just as fast (as discussed above) as merged dense models and more memory-efficient. In that regard, you could (loosely speaking) consider this as an alternative PTQ method.
>
> Based on this potential of LQ-LoRA as an alternative to PTQ, we performed additional experiments by using LQ-LoRA as a potential alternative to PTQ methods by training on a larger calibration dataset. Specifically, we train the LQ-LoRA (Fisher, 2.75-bit, 64-rank) for one epoch on two C4 partitions and WikiText-2. We then further quantize the LoRA components themselves to 8-bit (NF8). Please see the Table below for the results, where the numbers other than LQ-LoRA are from OmniQuant [1]. Note that we use the term "effective bits" to denote bits per parameter that treat the LoRA components as "overheads" for comparing with methods without LoRA; LoRA components (with NF-8 quantization) amount to about 0.1-0.2 extra bits per parameter.
>
> | Base Model | Method                           | Effective Bits (7B, 70B) | C4 (7B) | C4 (70B) | WikiText2 (7B) | WikiText2 (70B) |
> | -------- | ---------------------------------- | ----------- | ---- | ---- | ---- | ---- |
> | LLaMA-2 | Dense                               | 16          | 6.97 | 5.52 | 5.47 | 3.31 |
> | LLaMA-2 | RTN                                 | 3.15        | 8.40 | 6.02 | 6.66 | 3.97 |
> | LLaMA-2 | GPTQ            | 3.15        | 7.89 | 5.85 | 6.29 | 3.85 |
> | LLaMA-2 | AWQ               | 3.15        | 7.84 | -    | 6.24 | -    |
> | LLaMA-2 | OmniQuant  | 3.15        | 7.75 | 5.85 | 6.03 | 3.78 |
> | LLaMA-2 | LQ-LoRA (Fisher)   | 2.95, 2.85 | 7.60 | 5.88 | 5.67 | 3.65 |
>
> [1] https://arxiv.org/abs/2308.13137
>
> These results indicate that LQ-LoRA can also be used as a state-of-the-art PTQ method. Given the promise of LQ-LoRA as a potential alternative to PTQ, we will include this comparison against PTQ results in the next iteration of the paper.
>
>
> **Novelty**
>
> *Similarly, I would like to raise a query about the paper's novelty. While this method undeniably enhances the current approach (Q-LoRA), from a PEFT perspective, there could be superior methods, particularly concerning inference challenges. On the topic of novelty, I await the insights of fellow reviewers.*
>
> We want to reiterate three main contributions on top of QLoRA:
> 1. We adopt a matrix decomposition view of initializing the quantized and low-rank components of LoRA adaptation. (Note that QLoRA does not perform explicit matrix decomposition)
> 2. Our ILP formulation enables us to search for the mixed-configuration quantization that fits specific resource requirements (e.g., 2.75 bits per parameter).
> 3. We extended the two methods above to a data-aware setting by incorporating the sensitivities of parameters.

---

> ### Author Response · Authors · 2023-11-21
> **Discussion**
>
> Hi there! Please let us know if the above response answered some of your questions in the original review, and please let us know if you have follow-up questions!
>
> In particular, we want to highlight a few key takeaways from our response:
> 1. The iterative algorithm adds a small extra computation head --- a fraction of the total training time.
> 2. We included GPTQ-LoRA comparisons but referred to them as "LREC" in the paper.
> 3. Inference with quantized + low-rank is not necessarily slower, especially in batch = 1 setting (per the latest releases from QLoRA; kudos to them).
> 4. Practitioners could use the "quantized + low-rank" models without merging. Such models could be just as fast as merged dense models and more memory-efficient. Based on this potential, we performed additional experiments by using LQ-LoRA as a potential alternative to PTQ methods by training on a larger calibration dataset.

---

> > ### Comment · Reviewer_CSBT · 2023-11-23
> >
> > Thank you for your detailed response. While my concerns about the paper have not been completely addressed, I am raising my score to 6 in the spirit of accepting other reviewers' perspectives on its novelty. I hope the paper will effectively address my concerns.

---

### Official Review · Reviewer_bc89 · 2023-11-01

**Soundness:** 4 excellent
**Presentation:** 3 good
**Contribution:** 3 good
**Rating:** 8
**Confidence:** 4

**Summary:**

This paper proposes LQ-LoRA, a method for fine-tuning LLMs in a memory-efficient manner. Each weight matrix is decomposed into a low-rank component and a quantized component. The paper makes three contributions relative to the previously proposed QLoRA paper (Dettmers et. al, 2023):
1) An iterative algorithm for initializing the quantized and low-rank components for approximating a weight matrix, to minimize the Frobenius approximation error.
2) An integer linear program for assigning the best quantization configuration to each weight matrix, under a specified total memory budget.
3) A data-aware quantization strategy, which assigns more weight during the matrix approximation to parameters that are more "important" according to the Fisher information matrix.

The paper shows that these methods yield meaningful improvements over baseline quantization methods, across experiments on (1) language modeling on C4 (Llama-2 model), (2) instruction tuning on OpenAssistant (Llama-2 model), and (3) fine-tuning on GLUE (RoBERTa-Large).

**Strengths:**

- Compressing a LLM by replacing each weight matrix into a quantized component (frozen) and low-rank component (which can be fine-tuned) is a great idea for attaining more memory efficient version(s) of a model.
- The proposed iterative initialization method (equation 2) is a natural and simple way to initialize the low-rank (16-bit) and quantized components for each weight matrix, that effectively reduces the approximation error of the method.
- The proposed methods give meaningful improvements over baselines (Table 2), across several tasks and model sizes.

**Weaknesses:**

- The idea of decomposing a weight matrix into a low-rank component and a quantized component had already been proposed in QLoRA (Dettmers et. al, 2023).
- A few ablations / baselines could be added, to make clearer where the gains of the method come from. For example:
1) How important is the ILP to LQ-LoRA? Can you show the performance of LQ-LoRA without the ILP?
2) Can you show the performance of the regular LoRA method (no quantization), and also quantization (at different bit-rates) without LoRA, in Table 2?
- I found it unusual that while the quantization configurations were optimized extensively (chosen via ILP), the rank of the low-rank components was kept fixed at 64 for the vast majority of experiments (except for Table 4). Perhaps the rank could also be chosen with the ILP?

**Questions:**

- Is the only difference between QLoRA+ILP, and LQ-LoRA, the initialization?
- Does the ILP budget, as well as the "bits per param" column, also consider the low-rank components?

Suggestions:
- Can you normalize the y-axes in Figure 1 to be relative error $||X - (Q + L1*L2)||_F / ||X||_F$, to make it easier to interpret?
- Perhaps a Table version of Figure 3 would be helpful to better see how much memory is taken by the low-rank vs. quantized components. Discussing this issue earlier on and more prominently would be helpful for giving readers a better intuitive understanding of what components of the system take the most memory.

---

> ### Author Response · Authors · 2023-11-16
> **Response**
>
> Thanks for the helpful review! Please take a look at our responses below.
>
> **Novelty with respect to QLoRA**
>
> We want to note that QLoRA does not perform explicit matrix decomposition. It performs quantization of the original weight matrix and learns additive low-rank updates. In contrast, we perform explicit matrix decomposition of the original matrix into low-rank and quantized components via a matrix reconstruction objective.
>
> **Ablations**
>
> *How important is the ILP to LQ-LoRA? Can you show the performance of LQ-LoRA without the ILP?*
>
> One of the main usages of ILP is to flexibly quantize the model to meet specific memory constraints (e.g., 2.75 bits/parameter). That being said, LQ-LoRA is effective even without the ILP. For example, Table 4 shows LQ-LoRA without ILP. Here, we use NF3 quantization configuration --- NF3 reuses the configuration of the QLoRA's NF4 quantization with 3-bit first-level quantization.
>
> *Can you show the performance of the regular LoRA method (no quantization), and also quantization (at different bit-rates) without LoRA, in Table 2?*
>
> We want to note that the QLoRA paper already demonstrated little performance loss between regular LoRA and QLoRA (4bit). Hence, we can think of QLoRA (4bit) performance as a proxy for regular LoRA.
>
> However, based on your suggestion/question, we performed additional experiments using LQ-LoRA as a potential alternative to PTQ methods by training on a larger calibration dataset. Specifically, we train the LQ-LoRA (Fisher, 2.75-bit, 64-rank) for one epoch on two C4 partitions and WikiText-2. We then further quantize the LoRA components themselves to 8-bit (NF8). Please see the Table below for the results, where the numbers other than LQ-LoRA are from OmniQuant [1]. Note that we use the term "effective bits" to denote bits per parameter that treat the LoRA components as "overheads" for comparing with methods without LoRA; LoRA components (with NF-8 quantization) amount to about 0.1-0.2 extra bits per parameter.
>
> | Base Model | Method                           | Effective Bits (7B, 70B) | C4 (7B) | C4 (70B) | WikiText2 (7B) | WikiText2 (70B) |
> | -------- | ---------------------------------- | ----------- | ---- | ---- | ---- | ---- |
> | LLaMA-2 | Dense                               | 16          | 6.97 | 5.52 | 5.47 | 3.31 |
> | LLaMA-2 | RTN                                 | 3.15        | 8.40 | 6.02 | 6.66 | 3.97 |
> | LLaMA-2 | GPTQ            | 3.15        | 7.89 | 5.85 | 6.29 | 3.85 |
> | LLaMA-2 | AWQ               | 3.15        | 7.84 | -    | 6.24 | -    |
> | LLaMA-2 | OmniQuant  | 3.15        | 7.75 | 5.85 | 6.03 | 3.78 |
> | LLaMA-2 | LQ-LoRA (Fisher)   | 2.95, 2.85 | 7.60 | 5.88 | 5.67 | 3.65 |
>
> [1] https://arxiv.org/abs/2308.13137
>
> These results indicate that LQ-LoRA can also be used as a state-of-the-art PTQ method. Given the promise of LQ-LoRA as a potential alternative to PTQ, we will include this comparison against PTQ results in the next iteration of the paper.
>
> **ILP for Rank Selection**
>
> This is a very good/interesting suggestion! We have looked into this by ILP-searching the bits and ranks together. One practical challenge is measuring the "storages/errors"  of the ranks of low-rank components. This is important to trade off bits of the quantized matrices and ranks of LoRA. Notice that LoRA components will be fine-tuned; searching the ranks using errors at initialization will unnecessarily favor more bits at the cost of lower ranks. We might be able to formulate this as a bi-level optimization problem (first searching the ranks and then searching the bits), but we will leave this as future work.
>
> **Questions**
>
> *Is the only difference between QLoRA+ILP, and LQ-LoRA, the initialization?*
>
> Yep!
>
> *Does the ILP budget, as well as the "bits per param" column, also consider the low-rank components?*
>
> This table does not take into account low-rank components since we use rank = 64 for our QLoRA and LREC (GPTQ + LoRA) baselines. However, the PTQ table above **does** take into account the contribution from the low-rank components to ensure fair comparison against other PTQ methods. We find that the LoRA components (if quantized) add about ~0.2 bits. We will clarify this further!
>
> **Suggestions**
>
> Thanks for the suggestions regarding the figures and tables! We will take these into account for the final version of the paper.

---

> > ### Comment · Reviewer_bc89 · 2023-12-04
> > **Thank your for your response**
> >
> > I want to thank the authors for their thoughtful response, and for the additional experiments they ran. I continue to believe that this paper makes a nice contribution, and keep my overall score (8) unchanged.

---

### Official Review · Reviewer_qudp · 2023-11-01

**Soundness:** 3 good
**Presentation:** 3 good
**Contribution:** 3 good
**Rating:** 8
**Confidence:** 4

**Summary:**

This work proposes a new initialization scheme for doing fine tuning of Large Language Models (LLMs) that have been subjected to Post-Training Quantization (PTQ). The authors motivate their problem by first discussing (along with appropriate references) that the conventional initialization scheme for LoRA, in which the first low-rank adapter is initialized as a Gaussian matrix, and the second low-rank factor is initialized to a zero-matrix, is suboptimal when fine-tuning a PTQ model. The initialization scheme proposed in this paper considers a low-rank + quantized decomposition of the LLM weight matrices. Subsequently, the low-rank factors are used as initializations for fine-tuning.

Most of the paper discusses how to obtain this quantized + low-rank decomposition of the matrix. They do so using an alternating minimization algorithm, wherein the the low-rank component is obtained by computing the SVD of the error residual from the quantized matrix, and the quantized component is obtained by quantizing the error residual from just the low-rank component. This alternating algorithm is a heuristic, and it is terminated when the objective function value, i.e., the Frobenius norm error between the original matrix and its quantized + low-rank decomposition starts diverging (or is small enough).

In addition to this, the work also considers a dynamic bit allocation strategy across different layers, and formulate this problem as an Integer Linear Program. This is a constrained optimization problem, which minimizes the Frobenius norm reconstruction error subject to a total target bit rate. Moreover, they also propose a data-aware quantization strategy, wherein instead of treating each parameter weight equivalently, their sensitivity with respect to the loss function is evaluated using Fisher matrix, and an alternative objective function is minimized instead.

The authors convincingly do extensive numerical evaluations on several tasks, and identify that the predominant regime where their initialization provides benefits is where aggressive compression is required (eg., sub 4-bit quantization bit requirement per parameter).

**Strengths:**

The work provides a new initialization strategy for fine-tuning LLMs that have been subjected to post-training quantization. Conventional LoRA initialization schemes fail are suboptimal for aggressive quantization regimes, and it is this regime where LQ-LoRA proposed in this work provides an advantage.

The simplicity of this approach is appealing, and it can be readily used with existing quantization schemes in addition to the NormalFloat (NF) quantization scheme utilized in this paper. Proposed ILP formulation of dynamic bit allocation and the data aware variant are also quite interesting. The comprehensive numerical evaluations are also quite descriptive, and clearly identifies where LQ-LoRA performs better, and where it does not.

**Weaknesses:**

I have some concerns in mind (which are not drawbacks of the paper), but it would be nice if the authors addressed and/or discussed them:

1. One of the contributions of this work is the ILP formulation for dynamic bit allocation across layers. This dynamic configuration of quantization parameters (e.g., bit-width, block size) subject to a total target memory budget is quite interesting. Hardware-wise, the proposed strategy necessitates mixed-precision compute (i.e., different bits for different layers). Even without the ILP, the $Q + L_1L_2$ decomposition requires handling $Q$ is low-precision format, whereas $L_1$ and $L_2$ is high (original)-precision format (eg., $16$-bit). Moreover, the ILP formulation outputs bit-budget allocation at quite fine resolutions like $2$ or $3$-bits. My concern is that such low precision is not easily available as current hardware primitives, i.e.,we can find a $4$ bit GPU, but can we find a $2$ bit GPU? I understand that the simulations are done in PyTorch that provide the flexibility of finer precisions (the authors also mention this on Page 5, "Implementation"). It would be worth discussing that this is a significant bottleneck in the deployment of this scheme for actual benefits in hardware. Please note again that I do not see this as a significant drawback of this paper, but it is important that the authors acknowledge this.

2. There is a recent work on joint low-rank and quantized decomposition of a matrix:

"Matrix Compression via Randomized Low Rank and Low Precision Factorization" (Saha, Srivastava & Pilanci) (https://arxiv.org/abs/2310.11028)

This work derives Frobenius norm error bounds for the low-rank decomposition of a matrix, in which the low-rank factors are also quantized. LQ-LoRA considers the low-rank factors to be in high-precision. This work is complementary in the sense that the the low-rank factors of the LQ-LoRA decomposition can also be quantized (in case, the hardware is limited to low-precision only). This will also help in circumventing the mixed-precision hardware issue mentioned in point 1 above (i.e., now $W$, $L_1$, and $L_2$ -- all three can be in the same precision). The analyses techniques proposed in this work can also be used to upper bound the Frobenius norm error of LQ-LoRA in order to make it more theoretically principled.

**Questions:**

I have a few questions:

1. The authors mention: "LoRA obviates the need to allocate memory for storing gradients and optimizer states" -- shouldn't it be stated as LoRA does not require us to store gradients for all parameters of the LLM, but only for the low-rank adapters, for which the number of parameters can be a fraction of the total number of LLM parameters?

2. Why did the authors choose NormalFloat instead of instead of a simple RTN quantization scheme? In principle, it seems that LQ-LoRA can be extended with any quantization scheme, if I'm not mistaken? And RTN has benefits over NF, such as no Gaussian modeling assumptions on the weights?

3. Where does the value $\delta = \frac{1}{2}\left(\frac{1}{30} + \frac{1}{32}\right)$ come from? Was it proposed in the NF paper? Does this value remain this same if more flexible quantization resolutions are considered (as is done in this paper in the ILP, but not probably in the NF paper)?

4. Page 3: How is NF quantization **lossless**?

5. In Fig. 1 (center), the weight matrices are just quantized using NF 3-bits, whereas on the right figure, a low-rank + quantized decomposition is obtained, where the quantization is again 3-bit. The caption says "LQ decomposition results in less quantization error". Is this a fair comparison in terms of the total memory requirement? Isn't it obvious that the right figure will have low error than center, since in the center, the residual from quantization error is approximated by a low-rank factorization in high-precision, whereas residual is not considered in the center?

6. In Alg. 1 pseudocode -- Does $B$ denote the quantization budget or the total number of blocks (as used in the main text)?

7. Page 6: What does it mean by: "weight matrix $\bf F$ has homogeneous rows or columns"? Please clarify in the main text.

8. Is Table 4 the data-aware or the agnostic variant? Also in Table 4: LQ-LoRA with rank $64$ on C4 has ppl $7.93$. unless I'm mistaken, shouldn't there be a corresponding $7.93$ value in Table 2 as well?

**Details Of Ethics Concerns:**

None.

---

> ### Author Response · Authors · 2023-11-15
> **Response**
>
> We thank the reviewer for their comments/questions. Please find our responses below.
>
> **Mixed Precision & Hardware**
>
> We note that our work is in the **weight only** quantization regime (as are QLoRA, GPTQ, etc.), where only the weights (and not the activations) are quantized to lower bits. The actual matmul is done in full precision after dequantization.
>
> Concretely, we use just-in-time de-quantize the (quantized) matrix $\mathbf{Q}$, execute matrix operations (e.g., `matmul(X, Q) ==> matmul(X, dequantize(Q))`), and throw away the de-quantized matrix. This supports arbitrary quantization, and the main technical difficulty is how to do de-quantization quickly. We outlined more details in the appendix.
>
> Although weight-only quantization cannot make use of (faster and more energy-efficient) lower-precision matmuls, weight-only quantization still has two practical benefits. First, at a macro level, having a smaller model (due to quantization) allows us to increase the batch size. This reduces the need for expensive data/model parallelism, which requires cross-device or even cross-node communication. Second, at a micro level, smaller quantized matrices allow us to reduce data movement between SRAM and HBM of the GPU. As an example, QLoRA's matrix multiplication implementation could be faster than dense matrix multiplication [1].
>
> [1] https://github.com/TimDettmers/bitsandbytes/releases
>
> **Related paper on Joint Low-rank and Quantized Matrix Decomposition**
>
> Thanks for the suggestion; this is an interesting paper! We will discuss this paper in the related works section.
>
> We primarily considered the cases in which $\mathbf{L}_1$ and $\mathbf{L}_2$ are floating-point parameters. This is necessary because we want to fine-tune these parameters. That being said, the techniques introduced in that paper paper might be helpful for other (future) use cases. For example, we could post-training merge $\mathbf{Q}, \mathbf{L}_1, \mathbf{L}_2$, and re-decompose them into three separate quantized matrices for inference.
>
> **Question 1**
>
> You are correct -- we will clarify this; thanks!
>
> **Question 2**
>
> You are correct in that the techniques in this paper are not specific to NF. QLoRA paper demonstrated the superior quality of such a scheme compared to classical integer quantization. Our preliminary experiments confirmed this, and hence, we chose NF.
>
> We conducted the following experiment to be more concrete about NF vs INT. We ran LQ (i.e., Section 3.1) on LLaMA-2 7B's dense parameters with `rank=64` and NF3/INT3/NF4/INT4 quantization configuration (i.e., `b0=3/4, b1=8, b2=fp32, B0=64, B1=256`), and measure the reconstruction errors (i.e., $||\mathbf{W} - (\mathbf{Q} + \mathbf{L}_1 \mathbf{L}_2) ||_2^2$). The table below shows that NF indeed performed better.
>
> | Method      | Reconstruction error (unit: $10^4$) |
> | ----------- | ----------- |
> | LQ with NF3      | 7.12       |
> | LQ with INT3     | 11.6        |
> | LQ with NF4      | 1.42       |
> | LQ with INT4     | 2.02        |
>
> **Question 3**
>
> This is the same $\delta$ in QLoRA, and we treated it as a constant for all settings. We will clarify further in the next iteration of the paper.
>
> **Question 4**
>
> We _defined_ $\mathbb{Q}$ as the set of matrices that can be quantized (Equation 1 is a minimization problem, not an exact decomposition problem). Thanks for pointing out the confusion; we will clarify this.
>
> **Question 5**
>
> The middle figure represents the setting of many (Q)LoRA methods, in which the Low-Rank components are initialized to be zero. In that sense, $\operatorname{quantize}(\mathbf{W}) + \mathbf{L}_1 \mathbf{L}_2 = \operatorname{quantize}(\mathbf{W})$ for those methods.
>
> **Question 6**
>
> Good catch -- it's meant for "budget" but shared similar symbols with $B_0$ and $B_1$.
>
> **Question 7**
>
> We say a matrix $\mathbf{F}$ has homogenous rows or columns when either rows or columns of the matrix have identical values. We will clarify this.
>
> **Question 8**
>
> Table 4 used the quantization configuration of NF3/NF4. Specifically, they use the block sizes and second-level quantization as the original NF4, but with the first-level quantization set to 3/4-bit. Table 2, however, used ILP to search for the quantization configuration, hence the difference.
>
> We made such decisions for Table 4 to remove the possible confounding variable of using ILP to search for the quantization configuration (i.e., we just wanted to see the effect of higher rank LoRA components). We will clarify!

---

> > ### Comment · Reviewer_qudp · 2023-11-23
> > **Acknowledgement**
> >
> > Thank you for your response!
> >
> > I'm not sure I understand the response to Qn. 4: How is NF quantization lossless?
> >
> > Nevertheless, I believe this is a good paper, and retain my score.

---

### Meta-Review · Area_Chair_CakE · 2023-12-11

**Metareview:**

This paper proposes LQ-LoRA, a method for fine-tuning LLMs in a memory-efficient manner. Each weight matrix is decomposed into a low-rank component and a quantized component. The paper makes three contributions:

* An iterative algorithm for initializing the quantized and low-rank components for approximating a weight matrix, to minimize the Frobenius approximation error.
* An integer linear program for assigning the best quantization configuration to each weight matrix, under a specified total memory budget.
* A data-aware quantization strategy, which assigns more weight during the matrix approximation to parameters that are more "important" according to the Fisher information matrix.

**Justification For Why Not Higher Score:**

* The idea of decomposing a weight matrix into a low-rank component and a quantized component had already been proposed in QLoRA (Dettmers et. al, 2023).
* A few ablations / baselines could be added, to make clearer where the gains of the method come from.

**Justification For Why Not Lower Score:**

* Compressing a LLM by replacing each weight matrix into a quantized component (frozen) and low-rank component (which can be fine-tuned) is a great idea for attaining more memory efficient version(s) of a model.
* The proposed iterative initialization method (equation 2) is a natural and simple way to initialize the low-rank (16-bit) and quantized components for each weight matrix, that effectively reduces the approximation error of the method.
* The proposed methods give meaningful improvements over baselines (Table 2), across several tasks and model sizes.

---

### Decision · Program_Chairs · 2024-01-16

Accept (poster)